# Revisiting Uncertainty Estimation for Node Classification: New Benchmark and Insights

## Abstract

Uncertainty estimation is an important task that can be essential for high-risk applications of machine learning. This problem is especially challenging for node-level prediction in graph-structured data, as the samples (nodes) are interdependent. However, there is no established benchmark that allows for the evaluation of node-level uncertainty estimation methods in a unified setup, covering diverse and meaningful distribution shifts. In this paper, we address this problem and propose such a benchmark, together with a technique for the controllable generation of data splits with various types of distribution shifts. Importantly, we describe the shifts that are specific to the graph-structured data. Our benchmark consists of several graph datasets equipped with various distribution shifts on which we evaluate the robustness of models and their uncertainty estimation performance. To illustrate the benchmark, we decompose the current state-of-the-art Dirichlet-based framework and perform an ablation study on its components. In our experiments on the proposed benchmark, we show that when faced with complex yet realistic distribution shifts, most models fail to maintain high classification performance and consistency of uncertainty estimates with prediction errors. However, ensembling techniques help to partially overcome significant drops in performance and achieve better results than distinct models.

## 1 Introduction

Uncertainty estimation is an important and challenging task with many applications in financial systems, medical diagnostics, autonomous driving, etc. It aims at quantifying the confidence of machine learning models and can be used to design more reliable decision-making systems. In particular, it enables one to solve such problems as misclassification detection, where the model has to assign higher uncertainty to the potential prediction errors, or out-of-distribution (OOD) detection, when the model is required to yield higher uncertainty for the samples from an unknown distribution. Depending on the source of uncertainty, it can be divided into *data uncertainty*, which describes the inherent noise in data due to the labeling mistakes or class overlap, and *knowledge uncertainty*, which accounts for insufficient amount of information for accurate predictions when the distribution of test data is different from the training one (Gal, 2016; Malinin, 2019).

The problem of uncertainty estimation for graph-structured data has recently started to gain attention. It is especially complex at the node level as one has to deal with interdependent samples that may come from different distributions, so their predictions can change significantly depending on the neighborhood. This problem has already been addressed in several studies, and the proposed methods are commonly based on the Dirichlet distribution and introduce various extensions to the Dirichlet framework (Sensoy et al., 2018; Malinin & Gales, 2018; Malinin, 2019; Charpentier et al., 2020), such as graph-based kernel Dirichlet estimation (Zhao et al., 2020) or graph propagation of Dirichlet parameters (Stadler et al., 2021).

However, the field of robustness and uncertainty estimation for node-level graph problems suffers from the absence of benchmarks with diverse and meaningful distribution shifts. Usually, the evaluation is limited to somewhat unrealistic distribution shifts, such as noisy node features (Stadler et al., 2021) or left-out classes (Zhao et al., 2020; Stadler et al., 2021). Importantly, Gui et al. (2022) try to overcome this issue and systematically construct a graph OOD benchmark, in which they explicitly make distinctions between covariate and concept shifts. However, the authors either consider

synthetic datasets or ignore the graph structure when creating distribution shifts. The problem with the mentioned approaches is that, in real applications, distribution shifts can be much more complex and diverse, and may depend on the global graph structure (for a more detailed discussion, refer to Appendix C). Thus, the existing benchmarks can be insufficient to reliably and comprehensively evaluate uncertainty estimation methods for graph-structured data. Therefore, the current status quo about the best uncertainty estimation methods for node classification remains unclear and requires further investigation.

In this work, we propose a new benchmark for evaluating robustness and uncertainty estimation in transductive node classification tasks. The main feature of our benchmark is a general approach to constructing the data splits with distribution shifts: it can be applied to any graph dataset, allows for generating shifts of different nature, and one can easily vary the sizes of splits. For demonstration purpose, we apply our method to 7 common node classification datasets and describe 3 particular strategies to induce distribution shifts. Using the proposed benchmark, we evaluate the robustness of various models and their ability to detect errors and OOD inputs. Thus, we show that the recently proposed Graph Posterior Network (Stadler et al., 2021) is consistently the best method for detecting the OOD inputs. However, the best results for the other tasks are achieved using Natural Posterior Networks (Charpentier et al., 2021). We also confirm that ensembling often allows one to improve the model performance — ensembles of GPNs achieve the best performance for OOD detection, while ensembles of NatPNs have the best predictive performance and error detection.

## 2 PROBLEM STATEMENT

We consider the problem of transductive node classification in an attributed graph $\mathcal{G} = (\mathbf{A}, \mathbf{X}, \mathbf{Y})$ with an adjacency matrix $\mathbf{A} \in \{0, 1\}^{n \times n}$, a node feature matrix $\mathbf{X} \in \mathbb{R}^{n \times d}$ and categorical targets vector $\mathbf{Y} \in \{1, \ldots, C\}^n$. We split the set of nodes $\mathcal{V}$ into several non-intersecting subsets depending on whether they are used for training, validation, or testing and if they belong to in-distribution (ID) or out-of-distribution (OOD) subset. Let $\mathbf{Y}_{\text{train}}$ denote the labels of train nodes $\mathcal{V}_{\text{train}}$. Given a graph $\mathcal{G}_{\text{train}} = (\mathbf{A}, \mathbf{X}, \mathbf{Y}_{\text{train}})$, we aim at predicting the labels $\mathbf{Y}_{\text{test}}$ of test nodes $\mathcal{V}_{\text{test}}$ and estimating the uncertainty measure $u_i \in \mathbb{R}$ associated with these predictions. The obtained uncertainty estimates are used to solve the misclassification detection and OOD detection problems.

## 3 PROPOSED BENCHMARK

This section describes our benchmark for evaluating uncertainty estimates and robustness to distribution shifts for node-level graph problems. The most important ingredient of our benchmark is a unified approach for the controllable generation of diverse distribution shifts that can be applied to any graph dataset. Our benchmark includes a collection of common node classification datasets, several data split strategies, a set of problems for evaluating robustness and uncertainty estimation performance, and the associated metrics. We describe these components below.

### 3.1 GRAPH DATASETS

While our approach can potentially be applied to any node classification or node regression dataset, for our experiments, we pick the following 7 datasets commonly used in the literature: 3 citation networks, including **CoraML**, **CiteSeer** (McCallum et al., 2000; Giles et al., 1998; Getoor, 2005; Sen et al., 2008) and **PubMed** (Namata et al., 2012), 2 co-authorship graphs — **CoauthorPhysics** and **CoauthorCS** (Shchur et al., 2018), and 2 co-purchase datasets — **AmazonPhoto** and **Amazon-Computers** (McAuley et al., 2015; Shchur et al., 2018).

### 3.2 DATA SPLITS

The most important ingredient of our benchmark is a general generating data splits in a graph $\mathcal{G}$ to yield non-trivial yet reasonable distribution shifts. For this purpose, we make a distinction between the ID parts that are described by $\mathrm{p}(\mathbf{Y}_{\text{in}}|\mathbf{X}, \mathbf{A})$ and shifted (OOD) parts where the targets may come from a significantly different distribution $\mathrm{p}(\mathbf{Y}_{\text{out}}|\mathbf{X}, \mathbf{A})$.

We define the following ID parts:

- `Train` contains the nodes $\mathcal{V}_{\text{train}}$ that are used for the regular training of models and represent the only observations that take part in gradient computation;

- `Valid-In` enables us to monitor the best model during the training stage by computing the validation loss for the nodes $\mathcal{V}_{\text{valid-in}}$ and choose the best checkpoint;

- `Test-In` is used for testing on the remaining in-distribution nodes $\mathcal{V}_{\text{test-in}}$ and represents the most simple setup that requires the model to reproduce in-distribution dependencies. Both `Valid-In` and `Test-In` parts are assumed to come from the exact same distribution as `Train`.

At the same time, we introduce the following OOD parts:

- `Valid-Out` contains the validation nodes $\mathcal{V}_{\text{valid-out}}$ that also can be used for monitoring but tends to be a more difficult part of graph with potentially different dependencies;

- `Test-Out` represents the most shifted part $\mathcal{V}_{\text{test-out}}$ and can be used for evaluating robustness of models to distribution shifts.

To construct a particular data split, we choose some characteristic $\sigma_i$ and compute it for every node $i \in \mathcal{V}$, as described in Section 3.3. This characteristic reflects some node property that may depend on features or graph structure. After that, we sort all nodes in ascending order of $\sigma_i$. Some fraction of nodes with the smallest values of $\sigma_i$ is considered to be ID, while the remaining ones become OOD and are split into `Valid-Out` and `Test-Out` based on their values of $\sigma_i$. Importantly, this general split strategy is very flexible — it allows one to vary the size of the training part and to analyze the effect of this size on the robustness and the quality of uncertainty estimates. The type of distribution shift depends on the choice of $\sigma_i$ and can also be easily varied.

In our experiments, we split the dataset in the following proportions. The half of the nodes with the smallest values of $\sigma_i$ are assumed to be ID and are split into `Train`, `Valid-In`, and `Test-In` uniformly at random in proportion 30%/10%/10%. The second half contains the remaining OOD nodes split into `Valid-Out` and `Test-Out` in the ascending order of $\sigma_i$ in proportion 10%/40%. As a result, the `Test-Out` part has the most significant distribution shift.

## 3.3 DISTRIBUTION SHIFTS

To define our data splits, it is necessary to choose some node characteristic $\sigma_i$ as a split factor. We aim to consider diverse characteristics which cover a variety of distribution shifts that may occur in practice. In a standard non-graph ML setup, shifts typically happen only in feature space (or, more generally, the joint distribution of features and targets may become shifted). In graph learning tasks, there can be shifts specifically related to the graph structure: the training part can be biased towards more popular nodes or may consist of nodes from a particular region in the graph. Thus, we consider the following representative data split strategies.

**Random** This is a standard approach to constructing the data splits, where the nodes are selected uniformly at random, i.e., we can take $\sigma_i$ to be a random position in a sorted list. This type of shift is not realistic for practical applications but can be helpful for the analysis: it shows how well the model generalizes given that the distribution does not change. The random splitting strategy also allows for evaluating the robustness of models when the size of the training dataset varies.

**Feature** This approach represents a family of possible feature-based shifts that do not take into account the graph structure explicitly. There are multiple ways to construct such shifts, e.g., a split can be based on values of one particular feature (Gui et al., 2022). However, to follow our general setup described above, we base a split on a continuous characteristic that can be computed for any dataset. Namely, we project the original features $\boldsymbol{x}_i \in \mathbb{R}^d$ into $\mathbb{R}^2$ via a random linear transform $\mathbf{W}$, where all entries $w_{ij}$ are independent and come from $\mathcal{N}(0, 1)$. After that, $\sigma_i$ is set to the distance between the node $i \in \mathcal{V}$ and the centroid of the projected data, so the most *central* nodes in terms of features are said to be ID, while OOD parts are close to *periphery*. This setup naturally corresponds to the situation when the training dataset consists of the most *typical* elements, while some outliers may be encountered at the inference stage. Thus, this type of shift tests the robustness of models to non-standard feature combinations.

We visualize all the split strategies applied to the **AmazonPhoto** dataset in Figure 1. The figures for the remaining datasets can be found in Appendix A. Here, one can see that the feature-based split does not introduce a notable structural shift, i.e., the nodes are distributed across all regions of the graph. This fact is additionally confirmed by our analysis in Appendix B: it is clear that there is no significant difference in the degree distribution and pairwise node distances between the ID and OOD parts. Our empirical observations confirm that the feature-based shifts are the easiest to handle by the considered methods.

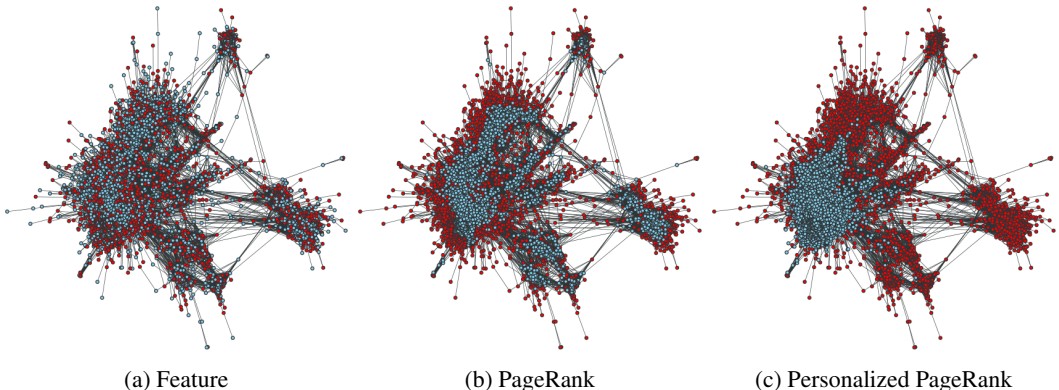

|  (a) Feature | (b) PageRank | (c) Personalized PageRank |

Figure 1: Visualization of data splits for **AmazonPhoto** dataset: ID is blue, OOD is red.

**PageRank**   This strategy represents a possible bias towards popularity. It is natural to expect the training set to consist of more popular items. For instance, in the web search, the importance of pages in the internet graph can be measured via PageRank (Page et al., 1999). For this application, the labeling of pages should start with important ones, since they are visited more often. Similar situations may happen for social networks, where it is natural to start labeling with the most influential users, or citation networks, where the most cited pages should be labeled first. However, when applying the model, it is essential to make accurate predictions on less popular elements. Motivated by that, we introduce a PageRank-based split. In particular, we compute the PageRank (PR) values for every node $i \in \mathcal{V}$ and define the measure $\sigma_i$ as the negative PR score, which means that the nodes with smaller values of PR (i.e., less important ones) come to the OOD subsets.

As can be seen in Figure 1, the PageRank-based split separates the most important nodes that belong to the cores of large clusters and the structural periphery, which consists of less important nodes in terms of PageRank. Our analysis in Appendix B confirms this observation: the degree distribution changes significantly across the ID and OOD subsets, tending to higher values for the ID nodes. The distance between such nodes also appears to be smaller on average. Our experiments prove that such a structural distribution shift creates a more severe challenge for the considered methods.

**Personalized PageRank**   This strategy is focused on a potential bias towards locality, which may happen when labeling is performed by exploring the graph starting from some node. For instance, this may occur in a web search where a crawler has to explore the web graph following the links. Similarly, information about the users of a social network can usually be obtained via an API, and new users are discovered following the friends of known users. To model such a situation, we use the concept of Personalized PageRank (PPR) (Page et al., 1999). It represents the stationary distribution of a random walk that always restarts from some fixed node (see, e.g., (Klicpera et al., 2018) for more details). The associated distribution shift naturally combines popularity and locality: PPR is related to node importance since the stationary distribution concentrates more on higher-degree nodes. On the other hand, the locality is also preserved since restarts always happen in a fixed node. For our splits, we select the node $j \in \mathcal{V}$ with the highest PR score as a restarting node and compute the PPR score for every node $i \in \mathcal{V}$. After that, we define the measure $\sigma_i$ as negative PPR. The nodes with high PPR, which belong to the ID part, are expected to be close to the restarting node, while far away nodes go to the OOD subset.

Figure 1 shows that locality is indeed preserved, as the ID part consists of one compact region around the most important node chosen as the restarting one. Thus, the ID subset includes periphery nodes as well as some nodes that were previously marked as the most important in the PR-based split but

are now less important for the restarting node. The remaining nodes come to the OOD part. Our analysis in Appendix B also provides strong evidence for the mentioned behavior: the PPR-based split strongly affects the distribution of pairwise distances within the ID/OOD parts as the locality bias of the ID part makes the OOD nodes even more distant from each other. The shift of the degree distribution is also notable but not as severe as for the PR-based split since here we consider only the popularity conditioned on some fixed node. Finally, our empirical results in Section 5 confirm that the PPR-based split is the most challenging one for graph neural networks.

## 3.4 METRICS

To evaluate the classification performance, we exploit standard *Accuracy* and compute these metrics on `Test-In` and `Test-Out` parts. To analyze the aggregated performance, we also report Accuracy and AUROC on the mixture of `Test-In` and `Test-Out`.

To evaluate the quality of uncertainty estimates, we consider two problems: error (misclassification) detection and OOD detection. To assess how well a model can detect misclassified samples, we use the concept of *Prediction Rejection Curve* (PRC) (Malinin et al., 2021; 2022). PRC traces the error rate as we replace model predictions with ground-truth labels in the order of decreasing uncertainty. If uncertainty is high for incorrectly classified samples, then the error rate is expected to drop quickly as we replace such predictions with ground true labels.

Thus, the *Area Under Prediction Rejection Curve* (AUPRC) evaluates the joint classification and uncertainty estimation performance, requiring the model not only to provide high prediction accuracy but also to signalize about possible errors through higher uncertainty scores. In our experiments, we compute $\text{AUPRC}_{\text{model}}$ on the merged test subset of nodes $\mathcal{V}_{\text{test}}$ using *total uncertainty* (TU), as the errors may occur due to the inherent noise in data or because of predicting on OOD samples.

A measure called *Prediction Rejection Ratio* (PRR) is also based on the Prediction Rejection Curve but evaluates only the ability of a model to detect misclassified samples. For this purpose, AUPRC is normalized as follows. Let $\text{PRC}_{\text{model}}$ be the predicted uncertainty estimates, $\text{PRC}_{\text{random}}$ be random uncertainty estimates, and $\text{PRC}_{\text{oracle}}$ be estimates that perfectly sort samples according to the prediction errors (i.e., all misclassified samples have higher oracle uncertainty). Then, the PRR metric is defined as follows:

$$\text{PRR} = \frac{\text{AUPRC}_{\text{random}} - \text{AUPRC}_{\text{model}}}{\text{AUPRC}_{\text{random}} - \text{AUPRC}_{\text{oracle}}}$$

The best value of this measure is 1 (for the perfect uncertainty estimates), while random uncertainty estimates give $\text{PRR} = 0$. Note that each model has its own oracle with the associated estimates that perfectly match its prediction errors. So the $\text{AUPRC}_{\text{oracle}}$ values of different models are independent and computed only based on the corresponding model predictions.

We also evaluate the ability of models to detect OOD samples. For this, we consider the mixture of `Test-In` and `Test-Out`. A good model is expected to have higher *knowledge uncertainty* (KU) (Gal, 2016; Malinin, 2019) values for the observations from `Test-Out` compared to `Test-In`. Here, we use the standard AUROC for the binary classification with positive events corresponding to the observations coming from the OOD subset.

## 4 METHODS

We consider several methods for estimating uncertainty in graph-related problems. Specifically, we cover message-passing neural networks, ensemble approaches Lakshminarayanan et al. (2017), and Dirichlet-based methods that are currently considered to be state-of-the-art for OOD detection that is called GPN (Stadler et al., 2021). For Dirichlet-based approaches, we conduct an ablation study to evaluate which design choices contribute most to performance.

### 4.1 STANDARD METHODS

In this class of methods, the constructed model $f_\theta$ predicts the parameters $\boldsymbol{\mu}_i = f_\theta(\boldsymbol{x}_i)^1$ of the categorical distribution $\mathrm{P}_\theta(y_i|\boldsymbol{x}_i) = \mathrm{P}(y_i|\boldsymbol{\mu}_i)$ in the standard classification task, while the uncertainty estimates are obtained based on the entropy of this distribution.

A simple baseline that serves us as a lower bound for our further experiments with more advanced methods is **MLP**, which represents a graph-agnostic MLP model and takes into account only the features of the current observation. Further, **GNN** is a simple GNN model based on a two-layer SAGE convolution (Hamilton et al., 2017), which combines the information from both the central node and its neighborhood. As a training objective, these methods use the standard *Cross-Entropy* loss between the one-hot-encoded target $y_i$ and the predicted categorical vector $\boldsymbol{\mu}_i$. For these methods, we can only define uncertainty as the entropy $u_i = \mathbb{H}\big[\mathrm{P}(y_i|\boldsymbol{\mu}_i)\big]$ of the predictive categorical distribution.

### 4.2 DIRICHLET-BASED METHODS

The core idea behind the Dirichlet-based uncertainty estimation methods is to model the pointwise Dirichlet distribution $\mathrm{p}_\theta(\boldsymbol{\mu}_i|\boldsymbol{x}_i) = \mathrm{p}(\boldsymbol{\mu}_i|\boldsymbol{\beta}_i^{\mathrm{post}})$ by predicting its parameters $\boldsymbol{\beta}_i^{\mathrm{feat}} = f_\theta(\boldsymbol{x}_i)$ and updating the uniform prior distribution with parameters $\boldsymbol{\beta}^{\mathrm{prior}}$ through their sum $\boldsymbol{\beta}_i^{\mathrm{post}} = \boldsymbol{\beta}_i^{\mathrm{feat}} + \boldsymbol{\beta}_i^{\mathrm{prior}}$. Using this Dirichlet distribution, one can obtain the target categorical one as follows:

$$\mathrm{P}_\theta(y_i|\boldsymbol{x}_i) = \mathbb{E}_{\mathrm{p}(\boldsymbol{\mu}_i|\boldsymbol{\beta}_i^{\mathrm{post}})}\mathrm{P}(y_i|\boldsymbol{\mu}_i).$$

It implies that $\mathrm{P}_\theta(y_i|\boldsymbol{x}_i)$ has parameters $\boldsymbol{\beta}_i^{\mathrm{post}}/S_i$, where $S_i = \sum_k \beta_{ik}^{\mathrm{post}}$ is called *evidence* or *precision*. In other words, the parameters of the categorical distribution can be obtained from the Dirichlet ones by normalization. Importantly, the Dirichlet-based methods allow us to distinguish between total and knowledge uncertainty as follows:

$$u_i^{\mathrm{total}} = \mathbb{H}\big[\mathrm{P}_\theta(y_i|\boldsymbol{x}_i)\big] = \mathbb{H}\big[\mathbb{E}_{\mathrm{p}(\boldsymbol{\mu}_i|\boldsymbol{\beta}_i^{\mathrm{post}})}\mathrm{P}(y_i|\boldsymbol{\mu}_i)\big], \quad u_i^{\mathrm{know}} = -S_i.$$

For this class of methods, the training objective is *Expected Cross-Entropy* with an optional regularisation term that is equal to the entropy of the predicted Dirichlet distribution $\mathrm{p}(\boldsymbol{\mu}_i|\boldsymbol{\beta}_i^{\mathrm{post}})$:

$$\mathcal{L}_i = \mathbb{E}_{\mathrm{p}(\boldsymbol{\mu}_i|\boldsymbol{\beta}_i^{\mathrm{post}})}\big[-\log\mathrm{P}(y_i|\boldsymbol{\mu}_i)\big] - \lambda\mathbb{H}\big[\mathrm{p}(\boldsymbol{\mu}_i|\boldsymbol{\beta}_i^{\mathrm{post}})\big]. \tag{1}$$

This loss function can be computed in closed form (Malinin & Gales, 2018; Charpentier et al., 2020).

As the most straightforward method in this class, we consider a modification of **GNN** that is referred to as **EN** (Evidential Network) (Sensoy et al., 2018) — while exploiting the same architecture, it is trained to predict the Dirichlet parameters via Loss (1).

There are also more advanced methods based on the Dirichlet distribution which induce the behavior of the underlying model by estimating the density function in the latent space using *Normalizing Flows* (Kingma et al., 2016; Huang et al., 2018). These methods can be united within the recently proposed framework *Posterior Networks* (Charpentier et al., 2020; 2021). It was applied to the node-level problems in (Stadler et al., 2021). In this paper, we provide a detailed study of this framework and consider different variations depending on how the density estimation is performed and how the graph information is used. The description of these components can be found in Appendix E.

### 4.3 ENSEMBLES

In our study, we also consider *ensembling* techniques that proved to increase the predictive performance of models and provide instruments for estimating uncertainty. Among the methods that predict the parameters $\boldsymbol{\mu}_i$ of categorical distributions $\mathrm{P}(y_i|\boldsymbol{\mu}_i)$, there is a widely used approach for uncertainty estimation introduced by Lakshminarayanan et al. (2017). It can be formulated as an

---

[1]For GNNs, we formally have $\boldsymbol{\mu}_i = f_\theta(\mathbf{A}, \mathbf{X}, i)$ but we write $f_\theta(\boldsymbol{x}_i)$ for simplicity of notation and consistency with non-graph methods.

empirical distribution of model parameters $\mathrm{q}(\theta|\mathcal{G}_{\text{train}})$ that can be obtained after training several instances of the model with different random seeds for initialization:

$$\mathrm{P}_{\theta}(y_i|\boldsymbol{x}_i) = \mathbb{E}_{\mathrm{q}(\theta|\mathcal{G}_{\text{train}})}\mathrm{P}(y_i|\boldsymbol{\mu}_i).$$

Given this, we can split the total uncertainty into data and knowledge uncertainty through the following expression (Malinin & Gales, 2018):

$$u_i^{\text{total}} = \mathbb{H}\big[\mathrm{P}_{\theta}(y_i|\boldsymbol{x}_i)\big] = \mathbb{H}\big[\mathbb{E}_{\mathrm{q}(\theta|\mathcal{G}_{\text{train}})}\mathrm{P}(y_i|\boldsymbol{\mu}_i)\big],$$

$$u_i^{\text{data}} = \mathbb{E}_{\mathrm{q}(\theta|\mathcal{G}_{\text{train}})}\mathbb{H}\big[\mathrm{P}(y_i|\boldsymbol{\mu}_i)\big], \quad u_i^{\text{know}} = u_i^{\text{total}} - u_i^{\text{data}}.$$

We apply this approach to **GNN** models and denote the obtained ensemble as **EnsGNN**. As for the Dirichlet-based approaches, we follow Charpentier et al. (2021) and define an ensemble of models that predict the parameters of posterior Dirichlet distribution as the mean over the parameters in ensemble. Here, uncertainty is estimated in the same way as for a single Dirichlet-based model.

## 5 EXPERIMENTS AND ABLATION

**Setup** As discussed in Section 3.4, we provide the comparison of methods in 4 different problems using their associated metrics. In particular, we report the standard Accuracy for general classification performance, PRR@TU for consistency of *total uncertainty* estimates $u_i^{\text{total}}$ with the prediction errors, AUPRC@TU for the joint classification and confidence performance, and AUROC@KU for the OOD detection using *knowledge uncertainty* estimates $u_i^{\text{know}}$. The details of our experimental setup can be found in Appendix G.

Our code and benchmark are publicly available at `https://anonymous.4open.science/r/revisiting-uncertainty-estimation-F4B6`, together with a framework for evaluating a variety of baseline models, Dirichlet-based methods, and ensembling techniques.

We run each method 5 times with different random seeds and report the mean and standard deviation. To compare a pair of methods and analyze whether one is noticeably better than the other, we report `win/tie/loss` counts aggregated over the datasets and, in some cases, over different distribution shifts. We say that a method wins on a particular dataset if it is better than its competitor and the difference is larger than the sum of their standard deviations. If the difference is smaller, there is a tie. Note that we do not compute the standard deviation for the ensembles since they combine all 5 models. To compare different methods, we aggregate their results over all the datasets as follows. First, for each dataset, metric, and distribution shift, we rank all the methods according to their performance (the smaller, the better). Then, we average the obtained ranks over all the datasets.

**Results and insights** In this section, we address several research questions. First, we analyze the complexity and diversity of the proposed data split strategies. Then, we demonstrate the performance of the considered methods on our benchmark. After that, we investigate the Dirichlet-based methods to figure out which of them is superior for each task. In conclusion, we find out whether the best-performing models can further be improved via ensembling.

*Q1: How complex and diverse are the proposed data splits?*

To answer this question, we analyze the predictive performance of the models on `Test-In` (ID) and `Test-Out` (OOD) parts. Table 1 shows this comparison for **GNN**, and similar results for **GPN**, **EnsGNN**, and **MLP** can be found in Tables 6–8 in Appendix. We can see that feature-based splits are the easiest for **GNN**: the difference between ID and OOD parts is sufficiently small, and the performance on the latter can be even better for some datasets. The PageRank splits are noticeably more complicated, but the most difficult split strategy is based on Personalized PageRank — the performance drop is dramatic on some datasets. This can be explained by the locality of the ID part (see Figure 1): the clear separation of ID from OOD in terms of the graph structure makes graph-based learning significantly more complicated.

*Q2: How do existing methods perform in our setup, and what is the current status quo?*

To answer this question, we compare the following models: **GNN** as the classic graph processing method, **EnsGNN** as the standard and universal approach allowing for getting uncertainty estimates,

Table 1: Accuracy of **GNN** on ID vs OOD test subsets. Diff.% is the difference between the accuracy scores on the OOD and ID parts divided by the accuracy score on the ID part.

| | **Feature** | | | **PageRank** | | | **PPR** | | |
|---|---|---|---|---|---|---|---|---|---|
| | ID | OOD | Diff, % | ID | OOD | Diff, % | ID | OOD | Diff, % |
| AmazonComputers | 84.88 | 84.86 | −0.02 | 87.86 | 82.36 | −6.27 | 86.40 | 58.77 | −31.97 |
| AmazonPhoto | 94.90 | 92.23 | −2.82 | 97.07 | 88.94 | −8.38 | 92.68 | 43.89 | −52.64 |
| CoauthorCS | 92.44 | 93.74 | +1.41 | 94.81 | 91.33 | −3.66 | 92.67 | 91.63 | −1.12 |
| CoauthorPhysics | 96.57 | 96.06 | −0.54 | 97.52 | 94.44 | −3.15 | 97.72 | 93.68 | −4.14 |
| CoraML | 83.68 | 86.96 | +3.92 | 89.90 | 82.89 | −7.80 | 86.76 | 75.51 | −12.96 |
| CiteSeer | 70.87 | 71.50 | +0.88 | 74.47 | 71.42 | −4.10 | 74.11 | 59.25 | −20.06 |
| PubMed | 87.58 | 86.01 | −1.79 | 88.69 | 83.39 | −5.98 | 85.88 | 84.53 | −1.57 |

Table 2: Average ranks of standard uncertainty estimation methods over graph datasets.

| | | Accuracy | PRR@TU | AUPRC@TU | AUROC@KU |
|---|---|---|---|---|---|
| **Random** | GNN | 2.3 | 1.7 | 2.3 | *n/a* |
| | GPN | 2.6 | 3.0 | 2.6 | *n/a* |
| | EnsGNN | 1.1 | 1.3 | 1.1 | *n/a* |
| **Feature** | GNN | 2.3 | 1.6 | 2.1 | 2.3 |
| | GPN | 2.6 | 2.9 | 2.7 | 1.0 |
| | EnsGNN | 1.1 | 1.6 | 1.1 | 2.7 |
| **PageRank** | GNN | 2.3 | 1.9 | 2.1 | 2.7 |
| | GPN | 2.4 | 2.7 | 2.7 | 1.0 |
| | EnsGNN | 1.3 | 1.4 | 1.1 | 2.3 |
| **PPR** | GNN | 2.4 | 2.3 | 2.6 | 2.9 |
| | GPN | 2.1 | 1.7 | 2.0 | 1.0 |
| | EnsGNN | 1.4 | 2.0 | 1.4 | 2.1 |

and **GPN** which is known to be state-of-the-art for the OOD detection (Stadler et al., 2021). Table 2 compares these methods over all datasets. One can see that, according to AUROC@KU, the best OOD detection performance is achieved by **GPN** for all types of shifts, which is consistent with previously reported results (Stadler et al., 2021). Unsurprisingly, **EnsGNN** shows the best predictive performance, as measured by Accuracy. Moreover, it has the most consistent uncertainty estimates in context of PRR@TU and provides the best joint performance via AUPRC@TU. In summary, ensembles are the best for all tasks but OOD detection, where the superior method is **GPN**.

*Q3: Which Dirichlet-based methods are the best for each prediction task?*

Here, we provide a detailed analysis of the Dirichlet-based framework. The simplest method is **EN**, which represents a standard GNN trained with the Loss (1). Further, we consider the methods using Normalizing Flows, and compare Standard vs Natural density estimation and Graph Encoding vs Graph Propagation. Based on Table 3, one can make the following conclusions. First, **GPN** is still the best method for OOD detection. Second, **NatPN** is the best according to all other tasks. Third, **EN** is a strong baseline that shows competitive results in Accuracy, PRR@TU, and AUCPRC@TU, often staying close to **NatPN**. To show whether the difference is statistically significant, we report the `win/tie/loss` counts for some pairs of models. See Table 4 for the aggregated results and Tables 9, 11–14 in Appendix for more details.

*Q4: Do ensembles consistently improve the performance of complex models?*

Ensemble are known to consistently improve the model performance in various tasks, so we aim to confirm that this result holds in the transductive node classification. For this purpose, we compare the ensembles of the two most promising methods **GPN** and **NatPN**. The aggregated results are shown in Table 5 (also, see Tables 10–14 in Appendix for more details). One can see that ensembling via **EnsGPN** consistently improves **GPN** for OOD detection, but the difference is mostly insignificant. In contrast, **EnsNatPN** is noticeably better than **NatPN**, and this gain is especially significant for Accuracy and the joint performance measured by AUPRC.

Table 3: Average ranks of Dirichlet-based methods over datasets, including **EN**, **PN**, **NatPN**, **GPN** and **NatGPN**, which are obtained for every considered split strategy and prediction task.

|          |        | Accuracy | PRR@TU | AUPRC@TU | AUROC@KU |
|----------|--------|----------|--------|----------|----------|
| **Random** | EN   | 2.0 | 1.7 | 2.0 | *n/a* |
|          | PN     | 3.7 | 3.7 | 3.4 | *n/a* |
|          | NatPN  | 1.9 | 1.4 | 1.9 | *n/a* |
|          | GPN    | 3.0 | 4.0 | 3.3 | *n/a* |
|          | NatGPN | 4.4 | 4.1 | 4.4 | *n/a* |
| **Feature** | EN  | 2.3 | 1.6 | 2.0 | 3.3 |
|          | PN     | 3.6 | 3.4 | 3.6 | 3.9 |
|          | NatPN  | 1.9 | 1.7 | 1.7 | 4.6 |
|          | GPN    | 2.9 | 4.4 | 3.3 | 1.4 |
|          | NatGPN | 4.4 | 3.9 | 4.4 | 1.9 |
| **PageRank** | EN | 2.1 | 2.0 | 2.1 | 4.4 |
|          | PN     | 3.9 | 3.4 | 3.6 | 3.1 |
|          | NatPN  | 1.7 | 1.9 | 1.7 | 4.4 |
|          | GPN    | 2.7 | 3.7 | 3.3 | 1.1 |
|          | NatGPN | 4.6 | 4.0 | 4.3 | 1.9 |
| **PPR**  | EN     | 3.0 | 3.0 | 2.9 | 3.3 |
|          | PN     | 3.7 | 3.9 | 3.9 | 3.4 |
|          | NatPN  | 2.0 | 1.7 | 1.9 | 5.0 |
|          | GPN    | 2.1 | 2.4 | 2.1 | 1.0 |
|          | NatGPN | 4.1 | 4.0 | 4.3 | 2.3 |

Table 4: `Win/tie/loss` counts for some pairs of Dirichlet-based methods across all the considered graph datasets and split strategies (except Random).

|               | Accuracy | PRR@TU | AUPRC@TU | AUROC@KU |
|---------------|----------|--------|----------|----------|
| NatPN vs EN   | 3/18/0   | 6/14/1 | 8/12/1   | 3/3/15   |
| GPN vs NatGPN | 14/5/2   | 8/11/2 | 15/2/4   | 12/8/1   |
| NatPN vs GPN  | 12/2/7   | 14/5/2 | 13/3/5   | 0/0/21   |

Table 5: `Win/tie/loss` counts for ensembles vs the corresponding single models across all the considered graph datasets and split strategies (except Random).

|                  | Accuracy | PRR@TU | AUPRC@TU | AUROC@KU |
|------------------|----------|--------|----------|----------|
| EnsGPN vs GPN    | 9/11/1   | 6/14/1 | 14/7/0   | 4/17/0   |
| EnsNatPN vs NatPN | 14/7/0  | 6/15/0 | 15/6/0   | 5/9/7    |

To summarize our findings, we refer to Table 15 for the comparison of all the methods in terms of ranks and to Tables 11–14 for the detailed comparison of their `win/tie/loss` counts. One can conclude that **GPN** is the best single-pass method for OOD detection, while **NatPN** is the best one for all other tasks. Moreover, the performance of the latter can be further improved by ensembling, so **EnsNatPN** achieves the best results in terms of Accuracy, PRR, and AUPRC.

## 6 CONCLUSION

In this work, we propose a new benchmark for evaluating robustness and uncertainty estimation in transductive node classification tasks. For this, we design a universal approach to creating data splits with distribution shifts: it can be applied to any graph dataset and allows for generating shifts of various nature. Using our benchmark, we show that the recently proposed Graph Posterior Network (Stadler et al., 2021) is consistently the best method for detecting the OOD inputs, while the best results for the other tasks are achieved using Natural Posterior Networks (Charpentier et al., 2021). Our experiments also confirm that ensembling allows one to improve the model performance. Thus, we believe that our benchmark will be useful for future studies of node-level uncertainty estimation.

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

# A  VISUALIZATION OF DISTRIBUTION SHIFTS

In Figures 2–8, we provide the visualization of different split strategies for all the datasets. Some graphs have multiple connected components — in that case, we keep only the largest one.

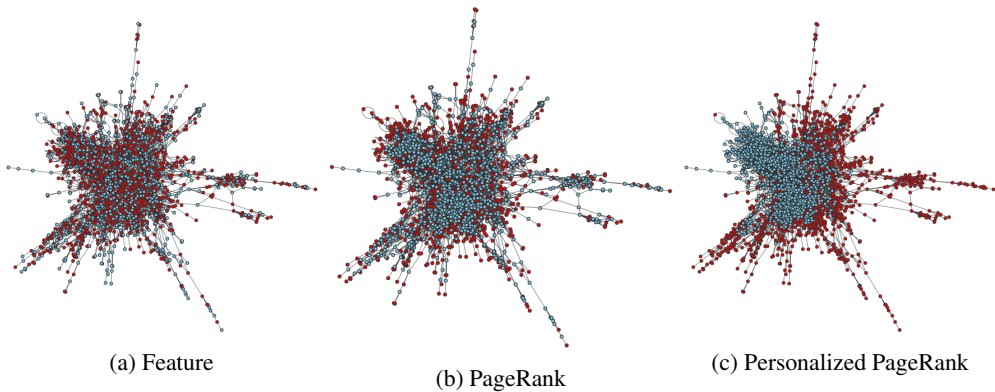

(a) Feature  (b) PageRank  (c) Personalized PageRank

Figure 2: Visualization of data splits for **CoraML** dataset: ID is blue, OOD is red.

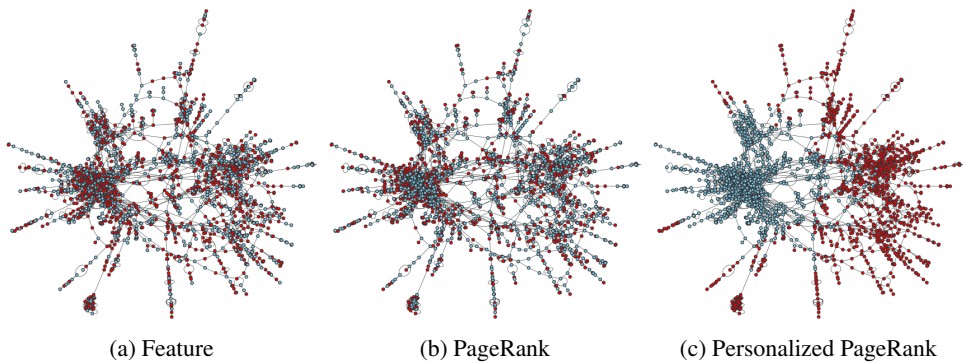

(a) Feature  (b) PageRank  (c) Personalized PageRank

Figure 3: Visualization of data splits for **CiteSeer** dataset: ID is blue, OOD is red.

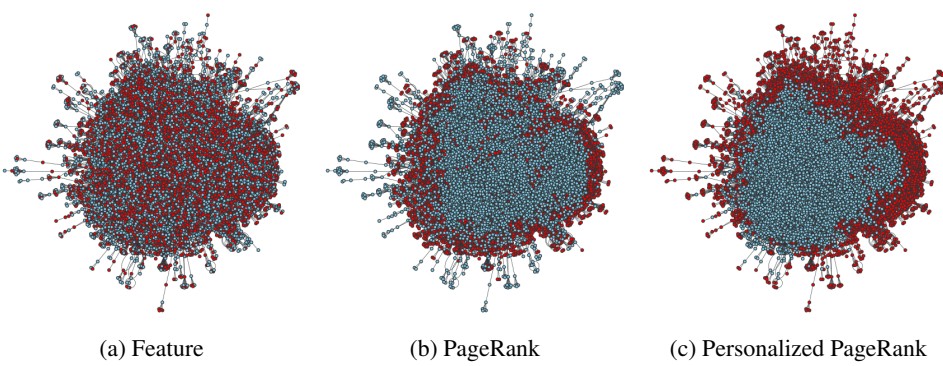

(a) Feature  (b) PageRank  (c) Personalized PageRank

Figure 4: Visualization of data splits for **PubMed** dataset: ID is blue, OOD is red.

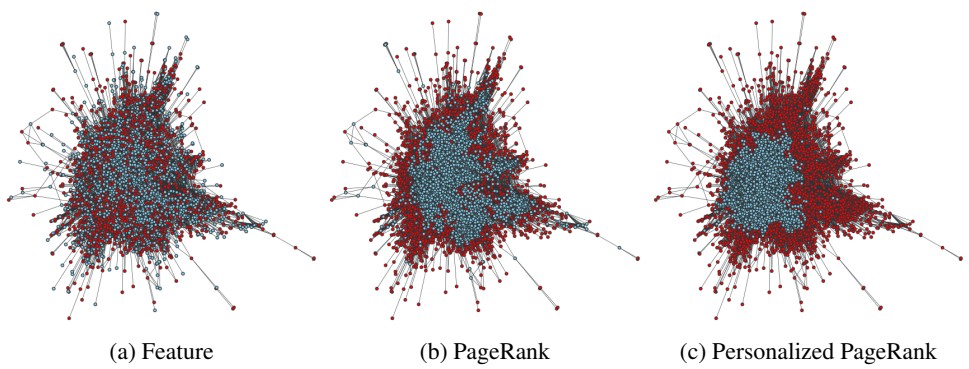

(a) Feature        (b) PageRank        (c) Personalized PageRank

Figure 5: Visualization of data splits for **AmazonComputers** dataset: ID is blue, OOD is red.

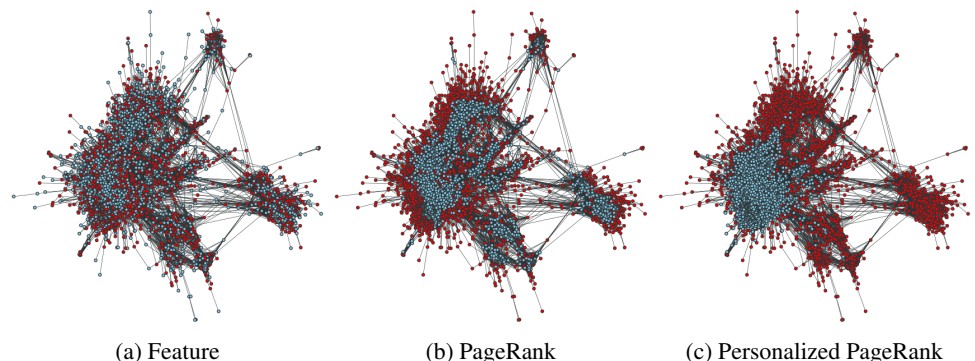

(a) Feature        (b) PageRank        (c) Personalized PageRank

Figure 6: Visualization of data splits for **AmazonPhoto** dataset: ID is blue, OOD is red.

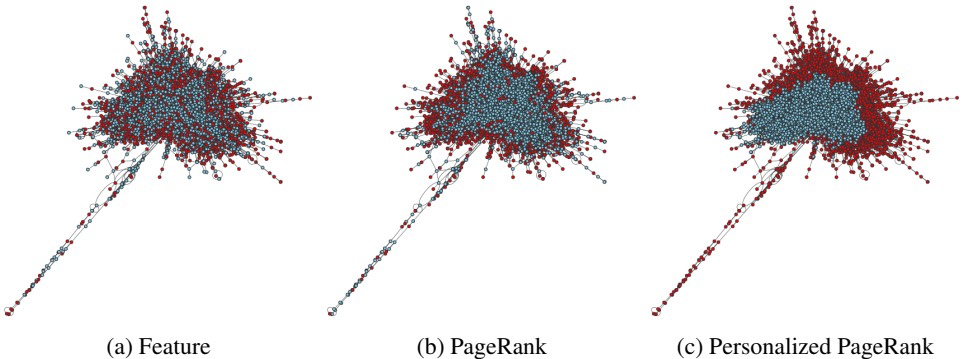

(a) Feature        (b) PageRank        (c) Personalized PageRank

Figure 7: Visualization of data splits for **CoauthorCS** dataset: ID is blue, OOD is red.

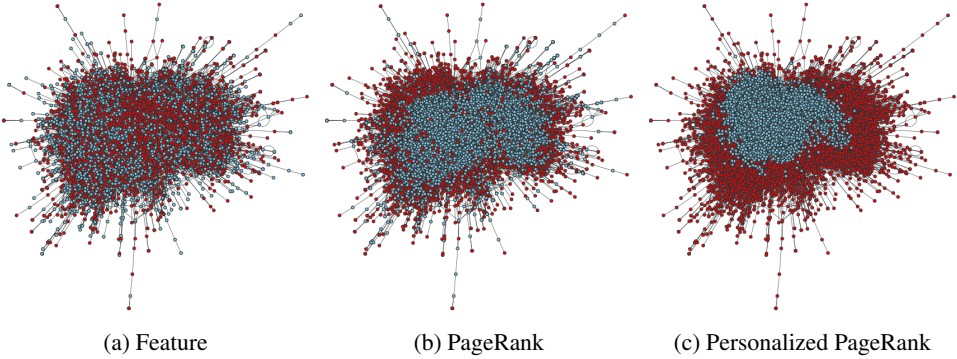

(a) Feature        (b) PageRank        (c) Personalized PageRank

Figure 8: Visualization of data splits for **CoauthorPhysics** dataset: ID is blue, OOD is red.

# B PROPERTIES OF DISTRIBUTION SHIFTS

This section provides a detailed analysis and comparison of the proposed distribution shifts. For this purpose, we consider three representative real-world datasets **AmazonComputers**, **CoauthorCS**, and **CoraML**, and discuss how different distribution shifts affect the basic properties of data: *class balance*, *degree distribution*, and *graph distances* between nodes within ID and OOD subsets.

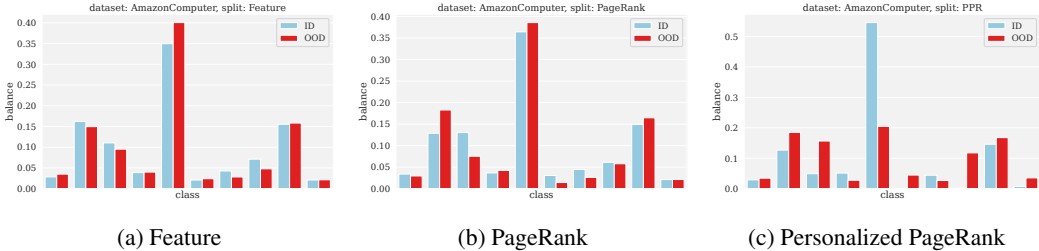

(a) Feature          (b) PageRank          (c) Personalized PageRank

Figure 9: Visualization of class balance for **AmazonComputers** dataset.

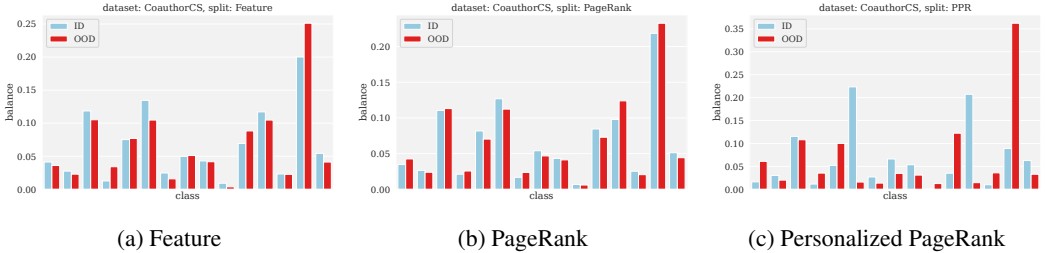

(a) Feature          (b) PageRank          (c) Personalized PageRank

Figure 10: Visualization of class balance for **CoauthorCS** dataset.

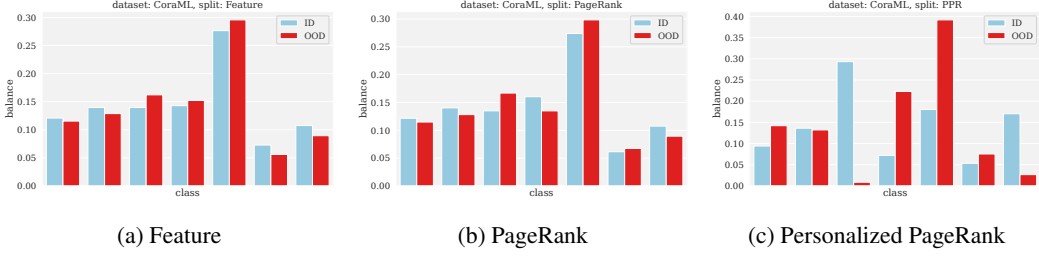

(a) Feature          (b) PageRank          (c) Personalized PageRank

Figure 11: Visualization of class balance for **CoraML** dataset.

**Class balance** Class balance directly affects the amount of evidence acquired by the graph processing model and used for estimating uncertainty and making predictions. It is especially important for evaluating Dirichlet-based models which exploit *Normalising Flows*, as their density estimates can become irrelevant due to a significant change in class balance.

In Figures 9–11, one can see that neither feature-based nor PageRank-based split makes a notable difference in the class balance between the ID and OOD subsets. At the same time, the PPR-based split leads to significant changes for some classes. This shows that the split strategies based on the structural locality in a graph can be very challenging as they also affect such crucial statistics as class balance. Interestingly, the PageRank-based split does not lead to significant shifts of class balance (for the datasets under consideration), i.e., the more important and less important nodes have, on average, the same probability of belonging to a particular class.

**Degree distribution** The node degree distribution is one of the basic structural characteristics of a graph that describes the local importance of nodes. Degrees are especially important for such graph

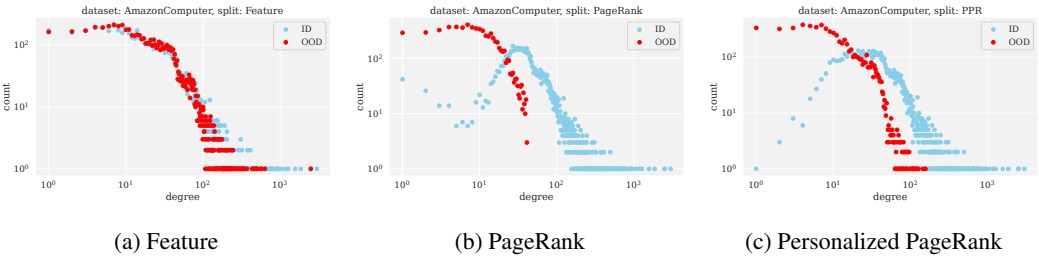

(a) Feature          (b) PageRank          (c) Personalized PageRank

Figure 12: Visualization of degree distribution for **AmazonComputers** dataset.

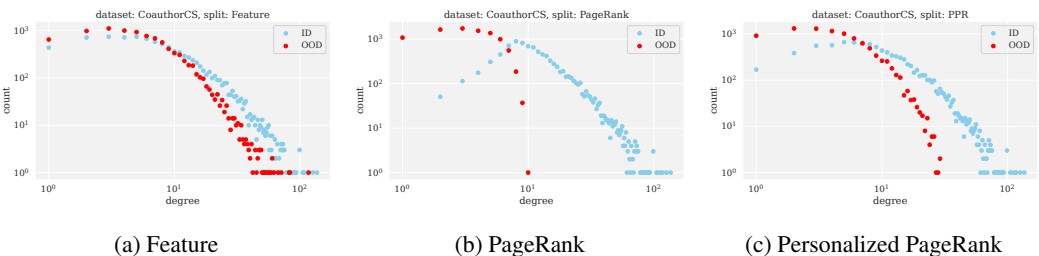

(a) Feature          (b) PageRank          (c) Personalized PageRank

Figure 13: Visualization of degree distribution for **CoauthorCS** dataset.

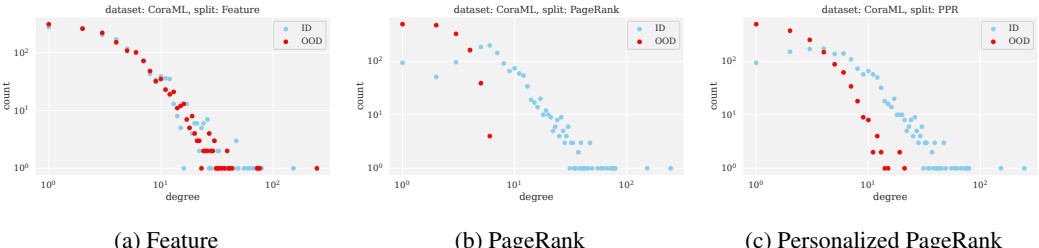

(a) Feature          (b) PageRank          (c) Personalized PageRank

Figure 14: Visualization of degree distribution for **CoraML** dataset.

processing methods as GNNs, since they describe how many channels around the considered node are used for message passing and aggregation.

In Figures 12–14, one can see that the most significant change in the degree distribution appears when the ID and OOD subsets are separated based on PageRank: the ID part contains more high-degree nodes. This is expected since PageRank is a graph characteristic measuring node importance (a.k.a. centrality), and node degree is the simplest centrality measure known to be correlated with PageRank. For PPR-based splits, the difference in degree distribution is smaller but still significant since PPR selects nodes by their relative importance for a particular node, so some high-degree nodes can be less important in terms of PPR. Finally, for features-based splits, the degree distribution also changes, but the shift level is much less significant.

**Distribution of pairwise distances** The distance between two nodes in a graph is defined as the length of the shortest path between them. Here, we compute such distances between the nodes in the ID or OOD subset within the original graph, i.e., we consider the whole graph when searching for the shortest path. The distribution of distances shows how easily messages can be passed between the nodes. Therefore, we expect that larger pairwise distances create more complicated tasks.

In Figures 15–17, one can observe that the PPR-based split leads to the most significant changes in distances, making the OOD nodes nearly twice as far from each other as the ID ones. At the same time, the PageRank-based split does not lead to such a difference, revealing almost the same distributions on ID and OOD subsets. This means that the popularity bias in a graph does not prevent one

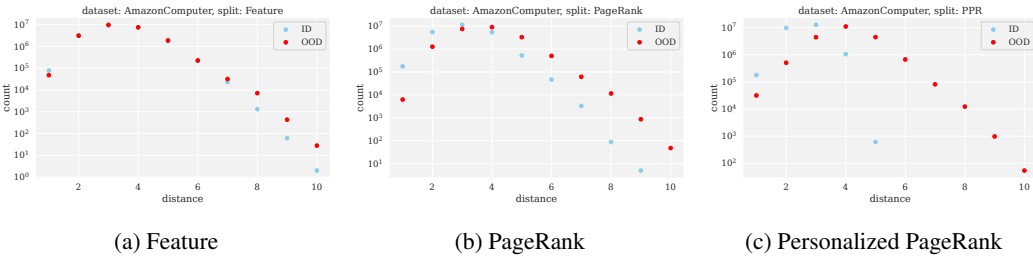

(a) Feature       (b) PageRank       (c) Personalized PageRank

Figure 15: Visualization of pairwise distance distribution for **AmazonComputers** dataset.

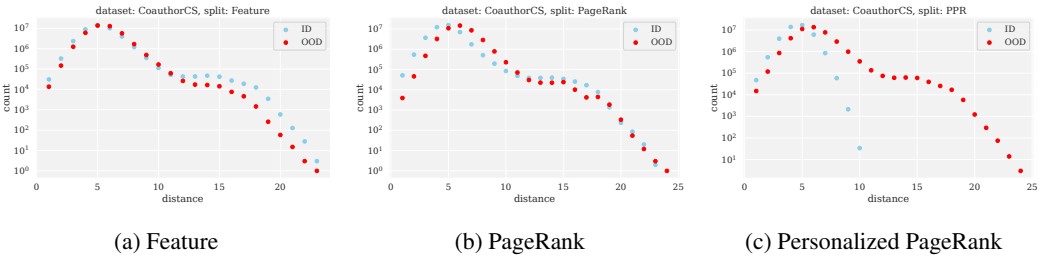

(a) Feature       (b) PageRank       (c) Personalized PageRank

Figure 16: Visualization of pairwise distance distribution for **CoauthorCS** dataset.

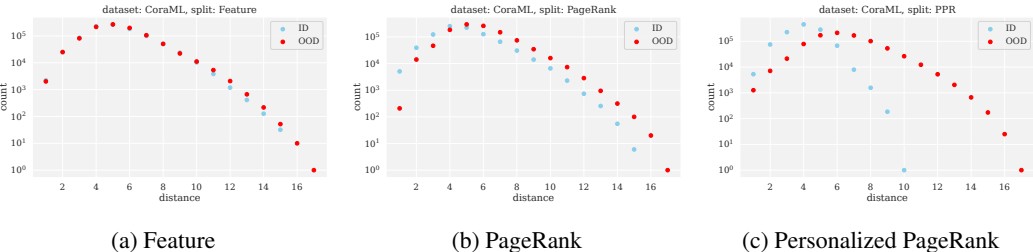

(a) Feature       (b) PageRank       (c) Personalized PageRank

Figure 17: Visualization of pairwise distance distribution for **CoraML** dataset.

from covering the less popular *periphery* nodes since the most popular nodes may be widespread. Finally, the feature-based split preserves the distances within the subsets.

## C    COMPARISON WITH GOOD BENCHMARK FROM GUI ET AL. (2022)

Our work complements and extends the GOOD benchmark recently proposed by Gui et al. (2022). However, there are several important differences that we discuss in this section.

One of the main properties of the GOOD benchmark is its theoretical distinction between two types of distribution shifts, which are represented through a graphical model. In particular, the authors consider covariate shifts, in which the distribution of features changes while the conditional distribution of targets given features remains the same, and concept shifts, where the opposite situation occurs, i.e., the conditional target distribution changes, while the feature distribution is the same. Although this distinction might be very helpful for understanding the properties of particular GNN models, such exclusively covariate or concept shifts rarely happen in practice where both types of shifts are present at the same time.

To create pure covariate or concept shifts, Gui et al. (2022) introduce different subsets of variables that either fully determine the target, create confounding associations with the target, or are completely independent of the target. This has to be properly handled and makes it non-trivial to create distribution shifts on new datasets with this approach. Indeed, the distribution shifts in the GOOD benchmark can be properly implemented only for synthetic graph datasets or via appending synthetic features that either describe various domains as completely independent variables or create

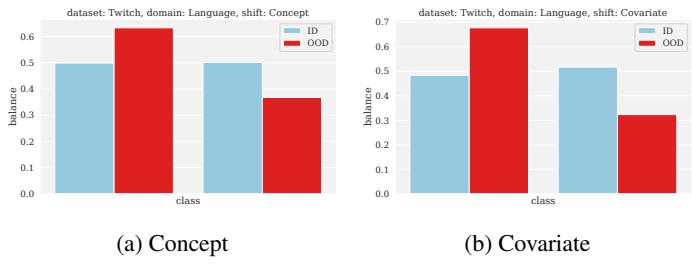

(a) Concept

(b) Covariate

Figure 18: Visualization of class balance for **GOOD-Twitch** dataset.

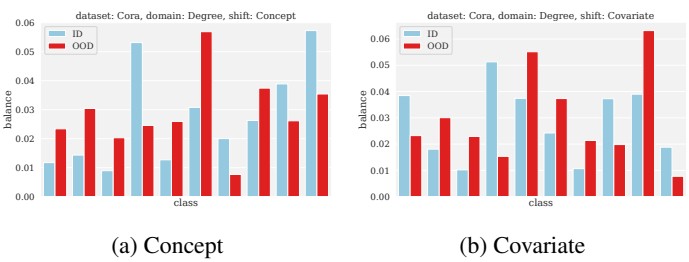

(a) Concept

(b) Covariate

Figure 19: Visualization of class balance for **GOOD-Cora** dataset.

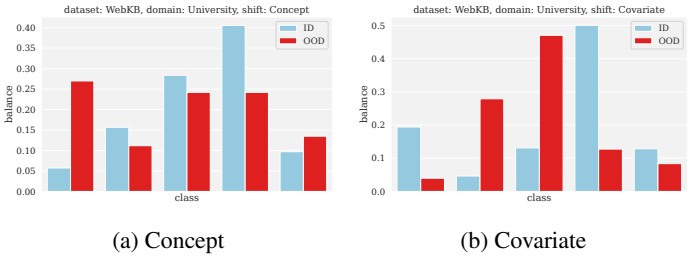

(a) Concept

(b) Covariate

Figure 20: Visualization of class balance for **GOOD-WebKB** dataset.

the necessary concepts by inducing some spurious correlation with the target. Moreover, the authors claim that, in the case of real-world datasets, one has to perform screening over the available node features to create the required setup of domain or concept shift. This fact implies numerous restrictions on how the data splits can be prepared.

In contrast, our method does not make a distinction between covariate and concept shifts and thus can be universally applied to any dataset and does not require any dataset modifications. Importantly, the type of distribution shift and the sizes of all split parts are easily controllable. This flexibility is the main advantage of our approach.

Finally, Gui et al. (2022) confirm the importance of using both node features and graph structure. Still, their node-level distribution shifts are usually based on node features such as the number of words or the year of publication in a citation network, the language of users in a social network, or the name of organizations in a webpage network. As for the graph properties, only node degrees are used in some citation networks. In contrast, we propose to use the graph structure directly and create significant distribution shifts using a very simple technique that requires computing some node property in the graph that should be chosen depending on a specific problem. For instance, one may use PageRank to create distribution shifts by the structural popularity of instances or Personalized PageRank to take into account the locality and distinguish between the core and periphery nodes.

Further in this section, we compare our benchmark with GOOD in terms of distribution shift statistics discussed in Appendix B.

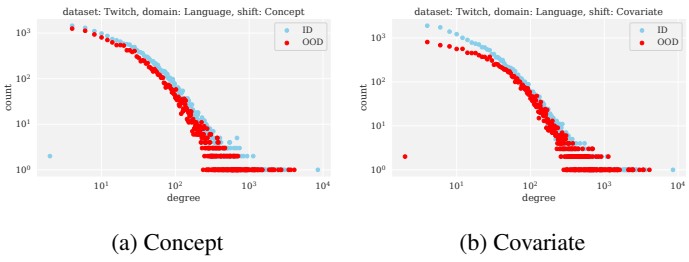

(a) Concept           (b) Covariate

Figure 21: Visualization of degree distribution for **GOOD-Twitch** dataset.

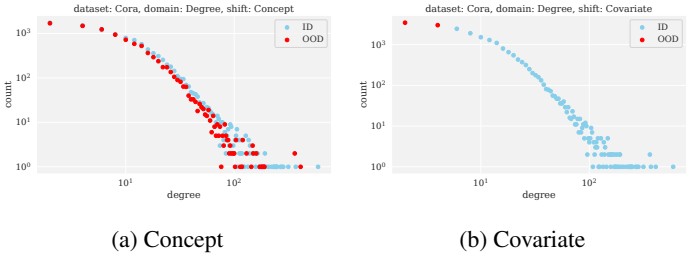

(a) Concept           (b) Covariate

Figure 22: Visualization of degree distribution for **GOOD-Cora** dataset.

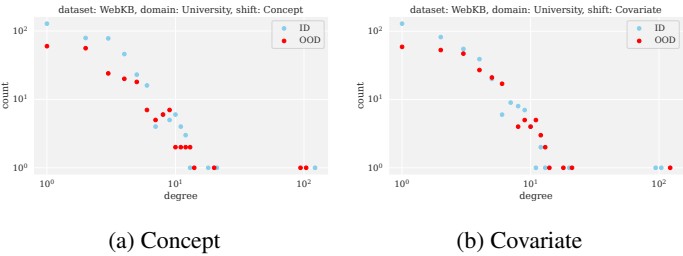

(a) Concept           (b) Covariate

Figure 23: Visualization of degree distribution for **GOOD-WebKB** dataset.

**Class balance** In our benchmark, the PPR-based shift significantly changes the class balance between ID and OOD subsets, while, for feature-based and PageRank-based shifts, the class balance does not change significantly. In GOOD, the level of distribution shift depends on the dataset and the type of shift. For instance, Figures 19 and 20 show a significant change for both types of shifts.

**Degree distribution** Comparing to our distribution shifts, the GOOD data splits have much less impact on the degree distribution: this graph property changes dramatically only when the covariate shift is constructed using the degree domain, as in the case of **GOOD-Cora** dataset (see Figure 22).

**Graph distance distribution** In contrast to our benchmark, the GOOD approach does not lead to a significant change in pairwise distance distribution between ID and OOD parts: the distance distribution hardly changes for concept shifts, and only covariate shifts make the difference between ID and OOD subsets somehow notable. This proves the necessity of considering the graph structure for inducing challenging distribution shifts.

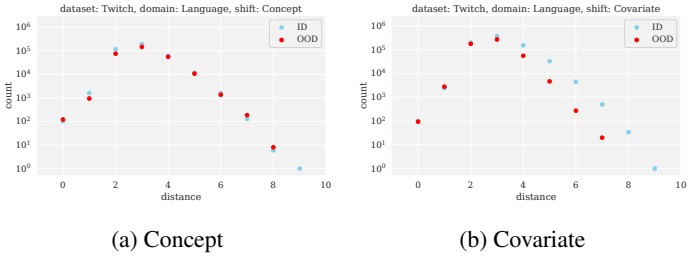

(a) Concept  (b) Covariate

Figure 24: Visualization of pairwise distance distribution for **GOOD-Twitch** dataset.

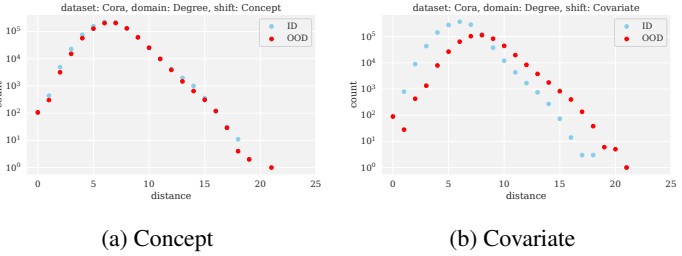

(a) Concept  (b) Covariate

Figure 25: Visualization of pairwise distance distribution for **GOOD-Cora** dataset.

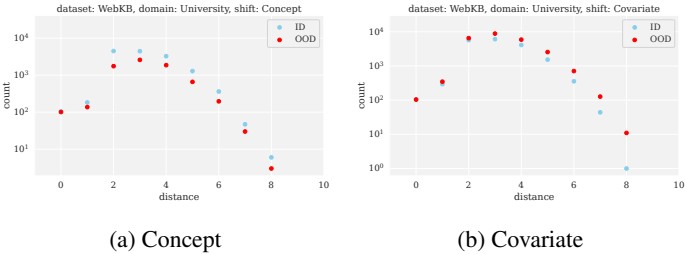

(a) Concept  (b) Covariate

Figure 26: Visualization of pairwise distance distribution for **GOOD-WebKB** dataset.

## D  ADDITIONAL RESULTS

In this section, we provide additional experimental results.

- Tables 6–8 are similar to Table 1 in the main text. They show how difficult our splits are for different models. These results are consistent: the feature-based split is the easiest for all the methods, while the PPR-based one is the hardest. Interestingly, it holds for the graph-agnostic **MLP**, which means that this graph-based shift also implies a noticeable shift in the feature space;

- Table 9 is a detalization of Table 4, where we consider different distribution shifts separately and add a random partition. Similarly, Table 10 detalizes Table 5;

- Tables 11–14 aggregates `win/tie/loss` counts for all pairs of methods;

- Table 15 compares all the methods in terms of their ranks averaged over the datasets;

- Tables 16–22 provide the results for all the methods on all the datasets. Note that all other aggregated results can be deduced from these tables.

Table 6: Accuracy of **MLP** on ID vs OOD test subsets for every split strategy.

| | Feature | | | PageRank | | | PPR | | |
| --- | --- | --- | --- | --- | --- | --- | --- | --- | --- |
| | ID | OOD | Diff, % | ID | OOD | Diff, % | ID | OOD | Diff, % |
| AmazonComputers | 75.80 | 77.69 | $+2.50$ | 81.09 | 75.49 | $-6.91$ | 81.18 | 50.23 | $-38.13$ |
| AmazonPhoto | 89.44 | 84.42 | $-5.61$ | 92.84 | 83.50 | $-10.06$ | 89.67 | 39.69 | $-55.74$ |
| CoauthorCS | 92.02 | 92.52 | $+0.54$ | 93.97 | 92.95 | $-1.09$ | 92.39 | 89.98 | $-2.61$ |
| CoauthorPhysics | 95.36 | 95.58 | $+0.23$ | 96.72 | 93.98 | $-2.84$ | 96.55 | 87.25 | $-9.63$ |
| CoraML | 75.18 | 70.78 | $-5.85$ | 75.18 | 68.88 | $-8.38$ | 72.04 | 52.50 | $-27.12$ |
| CiteSeer | 65.35 | 64.54 | $-1.24$ | 68.35 | 65.36 | $-4.37$ | 68.17 | 52.37 | $-23.18$ |
| PubMed | 85.29 | 84.50 | $-0.93$ | 86.15 | 85.10 | $-1.21$ | 84.43 | 84.04 | $-0.46$ |

Table 7: Accuracy of **GPN** on ID vs OOD test subsets for every split strategy.

| | Feature | | | PageRank | | | PPR | | |
| --- | --- | --- | --- | --- | --- | --- | --- | --- | --- |
| | ID | OOD | Diff, % | ID | OOD | Diff, % | ID | OOD | Diff, % |
| AmazonComputers | 88.05 | 89.19 | $+1.30$ | 89.83 | 83.58 | $-6.95$ | 88.63 | 65.84 | $-25.71$ |
| AmazonPhoto | 92.89 | 91.32 | $-1.69$ | 94.77 | 87.48 | $-7.69$ | 89.78 | 65.25 | $-27.32$ |
| CoauthorCS | 89.02 | 91.06 | $+2.29$ | 92.14 | 87.99 | $-4.51$ | 89.80 | 87.93 | $-2.08$ |
| CoauthorPhysics | 95.45 | 95.03 | $-0.44$ | 96.31 | 92.68 | $-3.77$ | 95.93 | 88.90 | $-7.32$ |
| CoraML | 84.01 | 84.72 | $+0.85$ | 86.96 | 83.11 | $-4.43$ | 84.48 | 75.81 | $-10.26$ |
| CiteSeer | 67.63 | 68.10 | $+0.70$ | 72.25 | 70.49 | $-2.44$ | 74.95 | 51.60 | $-31.16$ |
| PubMed | 88.95 | 86.38 | $-2.89$ | 88.45 | 85.46 | $-3.37$ | 86.12 | 86.03 | $-0.10$ |

Table 8: Accuracy of **EnsGNN** on ID vs OOD test subsets for every split strategy.

| | Feature | | | PageRank | | | PPR | | |
| --- | --- | --- | --- | --- | --- | --- | --- | --- | --- |
| | ID | OOD | Diff, % | ID | OOD | Diff, % | ID | OOD | Diff, % |
| AmazonComputers | 86.63 | 85.91 | $-0.83$ | 88.08 | 83.53 | $-5.17$ | 86.63 | 59.97 | $-30.77$ |
| AmazonPhoto | 95.16 | 92.68 | $-2.61$ | 97.65 | 89.77 | $-8.07$ | 92.68 | 45.03 | $-51.41$ |
| CoauthorCS | 92.58 | 94.00 | $+1.53$ | 94.65 | 91.40 | $-3.44$ | 92.80 | 91.86 | $-1.01$ |
| CoauthorPhysics | 96.52 | 96.07 | $-0.47$ | 97.56 | 94.49 | $-3.15$ | 97.74 | 93.79 | $-4.04$ |
| CoraML | 84.62 | 87.48 | $+3.38$ | 90.30 | 83.31 | $-7.75$ | 87.63 | 77.05 | $-12.07$ |
| CiteSeer | 72.07 | 72.50 | $+0.60$ | 74.77 | 71.90 | $-3.84$ | 74.77 | 61.91 | $-17.21$ |
| PubMed | 88.64 | 86.78 | $-2.10$ | 89.20 | 84.06 | $-5.76$ | 86.66 | 85.61 | $-1.22$ |

Table 9: `Win/tie/loss` counts for some pairs of Dirichlet-based methods across all the considered graph datasets.

| | | Accuracy | PRR@TU | AUPRC@TU | AUROC@KU |
| --- | --- | --- | --- | --- | --- |
| NatPN vs EN | Random | 0/7/0 | 1/6/0 | 2/4/1 | $n/a$ |
| | Feature | 1/6/0 | 2/5/0 | 2/4/1 | 1/1/5 |
| | PageRank | 1/6/0 | 1/6/0 | 3/4/0 | 2/1/4 |
| | PPR | 1/6/0 | 3/3/1 | 3/4/0 | 0/1/6 |
| GPN vs NatGPN | Random | 4/2/1 | 1/5/1 | 4/2/1 | $n/a$ |
| | Feature | 5/1/1 | 1/6/0 | 5/0/2 | 2/5/0 |
| | PageRank | 5/2/0 | 3/3/1 | 5/1/1 | 5/1/1 |
| | PPR | 4/2/1 | 4/2/1 | 5/1/1 | 5/2/0 |
| NatPN vs GPN | Random | 5/0/2 | 6/1/0 | 5/1/1 | $n/a$ |
| | Feature | 4/1/2 | 7/0/0 | 5/1/1 | 0/0/7 |
| | PageRank | 5/0/2 | 5/1/1 | 5/1/1 | 0/0/7 |
| | PPR | 3/1/3 | 2/4/1 | 3/1/3 | 0/0/7 |

Table 10: `Win/tie/loss` counts for ensembles vs the corresponding single models across all the considered graph datasets.

|  |  | Accuracy | PRR@TU | AUPRC@TU | AUROC@KU |
|---|---|---|---|---|---|
| EnsGPN vs GPN | Random | 6/1/0 | 1/4/2 | 6/1/0 | $n/a$ |
|  | Feature | 2/4/1 | 3/4/0 | 6/1/0 | 0/7/0 |
|  | PageRank | 4/3/0 | 1/5/1 | 4/3/0 | 4/3/0 |
|  | PPR | 3/4/0 | 2/5/0 | 4/3/0 | 0/7/0 |
| EnsNatPN vs NatPN | Random | 3/4/0 | 2/5/0 | 4/3/0 | $n/a$ |
|  | Feature | 3/4/0 | 2/5/0 | 5/2/0 | 1/3/3 |
|  | PageRank | 5/2/0 | 3/4/0 | 6/1/0 | 3/2/2 |
|  | PPR | 6/1/0 | 1/6/0 | 4/3/0 | 1/4/2 |

Table 11: Pairwise `win/tie/loss` counts for Accuracy across all graph datasets and split strategies (except Random).

|  | MLP | GNN | EN | PN | NatPN | GPN | NatGPN | EnsGNN | EnsNatPN | EnsGPN |
|---|---|---|---|---|---|---|---|---|---|---|
| MLP | 0/21/0 | 2/0/19 | 2/1/18 | 10/2/9 | 2/1/18 | 5/1/15 | 6/6/9 | 2/0/19 | 2/0/19 | 5/0/16 |
| GNN | 19/0/2 | 0/21/0 | 1/20/0 | 17/1/3 | 0/18/3 | 12/2/7 | 16/5/0 | 0/6/15 | 1/5/15 | 13/0/8 |
| EN | 18/1/2 | 0/20/1 | 0/21/0 | 16/1/4 | 0/18/3 | 12/3/6 | 14/7/0 | 0/7/14 | 0/6/15 | 12/1/8 |
| PN | 9/2/10 | 3/1/17 | 4/1/16 | 0/21/0 | 3/2/16 | 0/8/13 | 10/5/6 | 3/1/17 | 3/1/17 | 1/3/17 |
| NatPN | 18/1/2 | 3/18/0 | 3/18/0 | 16/2/3 | 0/21/0 | 12/2/7 | 17/4/0 | 3/6/12 | 0/7/14 | 12/2/7 |
| GPN | 15/1/5 | 7/2/12 | 6/3/12 | 13/8/0 | 7/2/12 | 0/21/0 | 14/5/2 | 6/1/14 | 6/1/14 | 1/11/9 |
| NatGPN | 9/6/6 | 0/5/16 | 0/7/14 | 6/5/10 | 0/4/17 | 2/5/14 | 0/21/0 | 0/1/20 | 0/1/20 | 4/2/15 |
| EnsGNN | 19/0/2 | 15/6/0 | 14/7/0 | 17/1/3 | 12/6/3 | 14/1/6 | 20/1/0 | 0/21/0 | 9/0/12 | 14/0/7 |
| EnsNatPN | 19/0/2 | 15/5/1 | 15/6/0 | 17/1/3 | 14/7/0 | 14/1/6 | 20/1/0 | 9/0/12 | 0/21/0 | 14/0/7 |
| EnsGPN | 16/0/5 | 8/0/13 | 8/1/12 | 17/3/1 | 7/2/12 | 9/11/1 | 15/2/4 | 7/0/14 | 7/0/14 | 0/21/0 |

Table 12: Pairwise `win/tie/loss` counts for PRR@TU across all graph datasets and split strategies (except Random).

|  | MLP | GNN | EN | PN | NatPN | GPN | NatGPN | EnsGNN | EnsNatPN | EnsGPN |
|---|---|---|---|---|---|---|---|---|---|---|
| MLP | 0/21/0 | 0/2/19 | 0/2/19 | 2/8/11 | 1/2/18 | 3/5/13 | 3/9/9 | 1/0/20 | 1/0/20 | 6/0/15 |
| GNN | 19/2/0 | 0/21/0 | 0/21/0 | 7/11/3 | 0/15/6 | 10/8/3 | 13/7/1 | 1/17/3 | 0/12/9 | 14/1/6 |
| EN | 19/2/0 | 0/21/0 | 0/21/0 | 8/10/3 | 1/14/6 | 10/7/4 | 14/6/1 | 7/8/6 | 3/8/10 | 12/4/5 |
| PN | 11/8/2 | 3/11/7 | 3/10/8 | 0/21/0 | 0/11/10 | 8/6/7 | 7/10/4 | 3/8/10 | 0/9/12 | 8/5/8 |
| NatPN | 18/2/1 | 6/15/0 | 6/14/1 | 10/11/0 | 0/21/0 | 14/5/2 | 14/6/1 | 8/7/6 | 0/15/6 | 15/3/3 |
| GPN | 13/5/3 | 3/8/10 | 4/7/10 | 7/6/8 | 2/5/14 | 0/21/0 | 8/11/2 | 5/3/13 | 2/4/15 | 1/14/6 |
| NatGPN | 9/9/3 | 1/7/13 | 1/6/14 | 4/10/7 | 1/6/14 | 2/11/8 | 0/21/0 | 1/4/16 | 1/4/16 | 5/5/11 |
| EnsGNN | 20/0/1 | 3/17/1 | 6/8/7 | 10/8/3 | 6/7/8 | 13/3/5 | 16/4/1 | 0/21/0 | 9/0/12 | 14/0/7 |
| EnsNatPN | 20/0/1 | 9/12/0 | 10/8/3 | 12/9/0 | 6/15/0 | 15/4/2 | 16/4/1 | 12/0/9 | 0/21/0 | 16/0/5 |
| EnsGPN | 15/0/6 | 6/1/14 | 5/4/12 | 8/5/8 | 3/3/15 | 6/14/1 | 11/5/5 | 7/0/14 | 5/0/16 | 0/21/0 |

Table 13: Pairwise `win/tie/loss` counts for AUPRC@TU across all graph datasets and split strategies (except Random).

|  | MLP | GNN | EN | PN | NatPN | GPN | NatGPN | EnsGNN | EnsNatPN | EnsGPN |
|---|---|---|---|---|---|---|---|---|---|---|
| MLP | 0/21/0 | 0/1/20 | 0/2/19 | 7/4/10 | 1/0/20 | 3/2/16 | 6/6/9 | 0/0/21 | 0/0/21 | 3/0/18 |
| GNN | 20/1/0 | 0/21/0 | 1/20/0 | 17/3/1 | 0/15/6 | 13/3/5 | 16/5/0 | 0/6/15 | 0/4/17 | 13/1/7 |
| EN | 19/2/0 | 0/20/1 | 0/21/0 | 16/1/4 | 1/12/8 | 13/3/5 | 18/3/0 | 0/4/17 | 0/2/19 | 13/0/8 |
| PN | 10/4/7 | 1/3/17 | 4/1/16 | 0/21/0 | 2/2/17 | 3/6/12 | 10/5/6 | 1/3/17 | 1/3/17 | 2/6/13 |
| NatPN | 20/0/1 | 6/15/0 | 8/12/1 | 17/2/2 | 0/21/0 | 13/3/5 | 17/4/0 | 6/2/13 | 0/6/15 | 13/2/6 |
| GPN | 16/2/3 | 5/3/13 | 5/3/13 | 12/6/3 | 5/3/13 | 0/21/0 | 15/2/4 | 4/2/15 | 5/0/16 | 0/7/14 |
| NatGPN | 9/6/6 | 0/5/16 | 0/3/18 | 6/5/10 | 0/4/17 | 4/2/15 | 0/21/0 | 0/2/19 | 0/1/20 | 5/0/16 |
| EnsGNN | 21/0/0 | 15/6/0 | 17/4/0 | 17/3/1 | 13/2/6 | 15/2/4 | 19/2/0 | 0/21/0 | 12/0/9 | 15/0/6 |
| EnsNatPN | 21/0/0 | 17/4/0 | 19/2/0 | 17/3/1 | 15/6/0 | 16/0/5 | 20/1/0 | 9/0/12 | 0/21/0 | 16/0/5 |
| EnsGPN | 18/0/3 | 7/1/13 | 8/0/13 | 13/6/2 | 6/2/13 | 14/7/0 | 16/0/5 | 6/0/15 | 5/0/16 | 0/21/0 |

Table 14: Pairwise `win/tie/loss` counts for AUROC@KU across all graph datasets and split strategies (except Random).

|          | MLP     | GNN     | EN      | PN      | NatPN   | GPN     | NatGPN  | EnsGNN  | EnsNatPN | EnsGPN  |
|----------|---------|---------|---------|---------|---------|---------|---------|---------|----------|---------|
| MLP      | 0/21/0  | 10/3/8  | 10/3/8  | 8/3/10  | 15/1/5  | 3/1/17  | 4/0/17  | 10/1/10 | 14/1/6   | 3/0/18  |
| GNN      | 8/3/10  | 0/21/0  | 1/18/2  | 6/6/9   | 15/3/3  | 0/3/18  | 1/3/17  | 5/4/12  | 15/0/6   | 0/1/20  |
| EN       | 8/3/10  | 2/18/1  | 0/21/0  | 5/7/9   | 15/3/3  | 0/3/18  | 1/6/14  | 7/4/10  | 15/1/5   | 0/1/20  |
| PN       | 10/3/8  | 9/6/6   | 9/7/5   | 0/21/0  | 17/3/1  | 0/1/20  | 1/3/17  | 10/3/8  | 17/1/3   | 0/0/21  |
| NatPN    | 5/1/15  | 3/3/15  | 3/3/15  | 1/3/17  | 0/21/0  | 0/0/21  | 0/3/18  | 4/1/16  | 7/9/5    | 0/0/21  |
| GPN      | 17/1/3  | 18/3/0  | 18/3/0  | 20/1/0  | 21/0/0  | 0/21/0  | 12/8/1  | 20/1/0  | 21/0/0   | 0/17/4  |
| NatGPN   | 17/0/4  | 17/3/1  | 14/6/1  | 17/3/1  | 18/3/0  | 1/8/12  | 0/21/0  | 17/2/2  | 19/1/1   | 3/6/12  |
| EnsGNN   | 10/1/10 | 12/4/5  | 10/4/7  | 8/3/10  | 16/1/4  | 0/1/20  | 2/2/17  | 0/21/0  | 15/0/6   | 0/0/21  |
| EnsNatPN | 6/1/14  | 6/0/15  | 5/1/15  | 3/1/17  | 5/9/7   | 0/0/21  | 1/1/19  | 6/0/15  | 0/21/0   | 0/0/21  |
| EnsGPN   | 18/0/3  | 20/1/0  | 20/1/0  | 21/0/0  | 21/0/0  | 4/17/0  | 12/6/3  | 21/0/0  | 21/0/0   | 0/21/0  |

Table 15: Average ranks of the considered methods across all datasets.

|          |          | Accuracy | PRR@TU | AUPRC@TU | AUROC@KU |
|----------|----------|----------|--------|----------|----------|
| **Random** | MLP      | 8.4      | 8.7    | 8.9      | $n/a$    |
|          | GNN      | 4.4      | 4.0    | 4.6      | $n/a$    |
|          | EN       | 4.6      | 3.6    | 4.9      | $n/a$    |
|          | PN       | 7.4      | 6.7    | 6.9      | $n/a$    |
|          | NatPN    | 4.6      | 3.0    | 4.4      | $n/a$    |
|          | GPN      | 6.6      | 7.7    | 6.9      | $n/a$    |
|          | NatGPN   | 8.7      | 8.1    | 8.7      | $n/a$    |
|          | EnsGNN   | 1.7      | 3.1    | 2.3      | $n/a$    |
|          | EnsNatPN | 3.0      | 2.3    | 1.9      | $n/a$    |
|          | EnsGPN   | 5.6      | 7.7    | 5.7      | $n/a$    |
| **Feature** | MLP      | 8.7      | 8.3    | 9.1      | 4.9      |
|          | GNN      | 4.0      | 4.3    | 4.4      | 5.7      |
|          | EN       | 5.4      | 2.6    | 4.7      | 6.3      |
|          | PN       | 7.4      | 6.4    | 7.0      | 7.1      |
|          | NatPN    | 4.7      | 3.3    | 4.4      | 8.0      |
|          | GPN      | 6.3      | 8.4    | 6.9      | 2.4      |
|          | NatGPN   | 8.4      | 7.3    | 8.4      | 3.4      |
|          | EnsGNN   | 1.7      | 4.3    | 2.0      | 6.9      |
|          | EnsNatPN | 3.0      | 2.7    | 2.1      | 8.1      |
|          | EnsGPN   | 5.3      | 7.4    | 5.9      | 2.1      |
| **PageRank** | MLP      | 6.9      | 9.0    | 7.9      | 6.4      |
|          | GNN      | 5.0      | 4.3    | 4.7      | 7.9      |
|          | EN       | 5.3      | 3.3    | 5.0      | 7.9      |
|          | PN       | 8.0      | 6.6    | 7.3      | 4.7      |
|          | NatPN    | 4.3      | 3.4    | 4.3      | 7.7      |
|          | GPN      | 6.0      | 7.0    | 7.0      | 1.9      |
|          | NatGPN   | 9.0      | 7.6    | 8.6      | 2.7      |
|          | EnsGNN   | 2.9      | 3.1    | 2.1      | 6.9      |
|          | EnsNatPN | 2.6      | 3.6    | 2.0      | 7.6      |
|          | EnsGPN   | 5.1      | 7.1    | 6.1      | 1.4      |
| **PPR**  | MLP      | 8.6      | 9.0    | 9.0      | 6.9      |
|          | GNN      | 5.1      | 5.3    | 5.6      | 6.1      |
|          | EN       | 6.4      | 5.6    | 6.1      | 5.6      |
|          | PN       | 7.4      | 7.3    | 7.6      | 6.4      |
|          | NatPN    | 4.9      | 3.7    | 4.3      | 9.0      |
|          | GPN      | 5.0      | 5.0    | 4.9      | 1.6      |
|          | NatGPN   | 8.4      | 7.6    | 8.4      | 3.7      |
|          | EnsGNN   | 3.0      | 4.9    | 3.1      | 5.0      |
|          | EnsNatPN | 2.4      | 3.1    | 2.6      | 9.3      |
|          | EnsGPN   | 3.7      | 3.6    | 3.4      | 1.4      |

Table 16: Experiment results on **AmazonComputers** dataset.

|          |          | Accuracy | PRR@TU | AUPRC@TU | AUROC@KU |
|----------|----------|----------|--------|----------|----------|
| **Random** | MLP | $78.44 \pm 0.57$ | $58.44 \pm 1.88$ | $88.32 \pm 0.27$ | $n/a$ |
|          | GNN | $86.33 \pm 0.56$ | $63.26 \pm 1.49$ | $93.80 \pm 0.27$ | $n/a$ |
|          | EN | $83.73 \pm 1.20$ | $67.42 \pm 3.81$ | $92.93 \pm 0.28$ | $n/a$ |
|          | PN | $88.18 \pm 0.23$ | $65.19 \pm 2.06$ | $94.98 \pm 0.22$ | $n/a$ |
|          | NatPN | $85.75 \pm 1.45$ | $65.22 \pm 3.38$ | $93.73 \pm 0.45$ | $n/a$ |
|          | GPN | $87.70 \pm 0.14$ | $59.35 \pm 0.25$ | $94.10 \pm 0.08$ | $n/a$ |
|          | NatGPN | $82.83 \pm 1.31$ | $56.45 \pm 2.69$ | $90.85 \pm 0.88$ | $n/a$ |
|          | EnsGNN | $86.67 \pm n/a$ | $64.22 \pm n/a$ | $94.09 \pm n/a$ | $n/a$ |
|          | EnsNatPN | $86.67 \pm n/a$ | $64.52 \pm n/a$ | $94.12 \pm n/a$ | $n/a$ |
|          | EnsGPN | $87.99 \pm n/a$ | $58.91 \pm n/a$ | $94.22 \pm n/a$ | $n/a$ |
| **Feature** | MLP | $77.31 \pm 0.41$ | $63.17 \pm 1.23$ | $88.39 \pm 0.17$ | $44.38 \pm 0.26$ |
|          | GNN | $84.87 \pm 1.08$ | $68.37 \pm 3.64$ | $93.66 \pm 0.23$ | $47.45 \pm 0.52$ |
|          | EN | $82.83 \pm 0.76$ | $72.01 \pm 2.81$ | $93.07 \pm 0.33$ | $47.38 \pm 0.40$ |
|          | PN | $88.39 \pm 0.50$ | $64.29 \pm 2.33$ | $94.98 \pm 0.44$ | $51.53 \pm 0.33$ |
|          | NatPN | $85.34 \pm 1.26$ | $65.52 \pm 4.06$ | $93.56 \pm 0.20$ | $51.19 \pm 0.96$ |
|          | GPN | $88.97 \pm 0.21$ | $56.81 \pm 1.16$ | $94.54 \pm 0.12$ | $53.33 \pm 0.31$ |
|          | NatGPN | $81.87 \pm 1.34$ | $63.70 \pm 6.12$ | $91.35 \pm 0.58$ | $53.46 \pm 0.57$ |
|          | EnsGNN | $86.05 \pm n/a$ | $67.15 \pm n/a$ | $94.11 \pm n/a$ | $48.36 \pm n/a$ |
|          | EnsNatPN | $86.29 \pm n/a$ | $64.76 \pm n/a$ | $93.95 \pm n/a$ | $51.92 \pm n/a$ |
|          | EnsGPN | $89.34 \pm n/a$ | $56.23 \pm n/a$ | $94.70 \pm n/a$ | $53.46 \pm n/a$ |
| **PageRank** | MLP | $76.61 \pm 0.42$ | $55.61 \pm 1.97$ | $86.58 \pm 0.27$ | $59.34 \pm 0.34$ |
|          | GNN | $83.46 \pm 0.62$ | $66.38 \pm 0.91$ | $92.62 \pm 0.35$ | $56.53 \pm 0.68$ |
|          | EN | $81.72 \pm 1.14$ | $68.72 \pm 2.71$ | $91.99 \pm 0.43$ | $56.32 \pm 1.33$ |
|          | PN | $84.70 \pm 0.69$ | $65.96 \pm 0.97$ | $93.25 \pm 0.37$ | $65.04 \pm 1.96$ |
|          | NatPN | $83.03 \pm 1.28$ | $66.84 \pm 2.34$ | $92.45 \pm 0.41$ | $48.01 \pm 2.04$ |
|          | GPN | $84.83 \pm 0.22$ | $59.26 \pm 1.69$ | $92.46 \pm 0.23$ | $90.58 \pm 0.49$ |
|          | NatGPN | $78.44 \pm 1.16$ | $57.22 \pm 3.61$ | $88.10 \pm 1.33$ | $87.31 \pm 2.33$ |
|          | EnsGNN | $84.44 \pm n/a$ | $65.98 \pm n/a$ | $93.11 \pm n/a$ | $61.00 \pm n/a$ |
|          | EnsNatPN | $84.44 \pm n/a$ | $65.73 \pm n/a$ | $93.08 \pm n/a$ | $44.68 \pm n/a$ |
|          | EnsGPN | $85.05 \pm n/a$ | $58.39 \pm n/a$ | $92.48 \pm n/a$ | $91.59 \pm n/a$ |
| **PPR** | MLP | $56.42 \pm 3.97$ | $54.18 \pm 3.29$ | $69.68 \pm 3.26$ | $79.23 \pm 0.76$ |
|          | GNN | $64.30 \pm 0.93$ | $70.18 \pm 3.24$ | $80.41 \pm 1.02$ | $79.69 \pm 1.07$ |
|          | EN | $63.44 \pm 0.84$ | $66.23 \pm 3.19$ | $78.79 \pm 1.23$ | $81.23 \pm 0.83$ |
|          | PN | $68.11 \pm 1.14$ | $61.39 \pm 4.21$ | $81.44 \pm 0.86$ | $69.79 \pm 2.28$ |
|          | NatPN | $64.10 \pm 1.26$ | $69.25 \pm 2.63$ | $80.03 \pm 1.21$ | $53.22 \pm 6.98$ |
|          | GPN | $70.40 \pm 0.31$ | $73.80 \pm 1.84$ | $85.78 \pm 0.30$ | $93.20 \pm 0.63$ |
|          | NatGPN | $58.25 \pm 4.36$ | $51.29 \pm 8.28$ | $70.60 \pm 5.99$ | $85.60 \pm 3.98$ |
|          | EnsGNN | $65.30 \pm n/a$ | $71.37 \pm n/a$ | $81.48 \pm n/a$ | $79.78 \pm n/a$ |
|          | EnsNatPN | $65.30 \pm n/a$ | $69.48 \pm n/a$ | $81.05 \pm n/a$ | $51.25 \pm n/a$ |
|          | EnsGPN | $70.77 \pm n/a$ | $74.65 \pm n/a$ | $86.21 \pm n/a$ | $93.38 \pm n/a$ |

Table 17: Experiment results on **AmazonPhoto** dataset.

|  |  | Accuracy | PRR@TU | AUPRC@TU | AUROC@KU |
|---|---|---|---|---|---|
| **Random** | MLP | $89.49 \pm 0.25$ | $73.44 \pm 0.65$ | $96.40 \pm 0.06$ | $n/a$ |
|  | GNN | $94.34 \pm 0.24$ | $83.81 \pm 1.38$ | $98.82 \pm 0.10$ | $n/a$ |
|  | EN | $94.25 \pm 0.19$ | $83.18 \pm 1.53$ | $98.76 \pm 0.10$ | $n/a$ |
|  | PN | $93.01 \pm 0.27$ | $81.37 \pm 1.28$ | $98.30 \pm 0.08$ | $n/a$ |
|  | NatPN | $94.04 \pm 0.15$ | $83.96 \pm 1.53$ | $98.75 \pm 0.05$ | $n/a$ |
|  | GPN | $92.67 \pm 0.13$ | $67.44 \pm 1.36$ | $97.25 \pm 0.08$ | $n/a$ |
|  | NatGPN | $89.09 \pm 4.56$ | $73.52 \pm 3.20$ | $96.10 \pm 2.29$ | $n/a$ |
|  | EnsGNN | $94.61 \pm n/a$ | $85.78 \pm n/a$ | $98.99 \pm n/a$ | $n/a$ |
|  | EnsNatPN | $94.51 \pm n/a$ | $84.87 \pm n/a$ | $98.91 \pm n/a$ | $n/a$ |
|  | EnsGPN | $92.92 \pm n/a$ | $67.50 \pm n/a$ | $97.36 \pm n/a$ | $n/a$ |
| **Feature** | MLP | $85.43 \pm 0.55$ | $72.02 \pm 1.41$ | $94.40 \pm 0.14$ | $57.38 \pm 0.27$ |
|  | GNN | $92.76 \pm 0.63$ | $80.34 \pm 1.63$ | $98.16 \pm 0.14$ | $59.36 \pm 0.39$ |
|  | EN | $91.60 \pm 0.57$ | $82.66 \pm 1.75$ | $97.96 \pm 0.22$ | $58.41 \pm 2.29$ |
|  | PN | $91.63 \pm 0.45$ | $77.16 \pm 2.06$ | $97.55 \pm 0.21$ | $57.29 \pm 1.10$ |
|  | NatPN | $92.15 \pm 0.33$ | $80.57 \pm 1.29$ | $97.98 \pm 0.16$ | $49.08 \pm 1.57$ |
|  | GPN | $91.63 \pm 0.37$ | $74.08 \pm 0.85$ | $97.31 \pm 0.09$ | $61.70 \pm 0.49$ |
|  | NatGPN | $87.83 \pm 1.31$ | $77.90 \pm 3.96$ | $96.12 \pm 0.91$ | $59.80 \pm 0.32$ |
|  | EnsGNN | $93.18 \pm n/a$ | $81.62 \pm n/a$ | $98.37 \pm n/a$ | $60.81 \pm n/a$ |
|  | EnsNatPN | $92.44 \pm n/a$ | $81.86 \pm n/a$ | $98.16 \pm n/a$ | $46.46 \pm n/a$ |
|  | EnsGPN | $91.76 \pm n/a$ | $75.26 \pm n/a$ | $97.45 \pm n/a$ | $61.75 \pm n/a$ |
| **PageRank** | MLP | $85.36 \pm 0.10$ | $66.95 \pm 0.59$ | $93.73 \pm 0.10$ | $62.74 \pm 0.25$ |
|  | GNN | $90.57 \pm 0.24$ | $75.05 \pm 0.75$ | $96.98 \pm 0.13$ | $63.09 \pm 0.77$ |
|  | EN | $90.33 \pm 0.37$ | $75.18 \pm 1.10$ | $96.90 \pm 0.13$ | $62.20 \pm 0.95$ |
|  | PN | $88.26 \pm 0.21$ | $73.56 \pm 1.40$ | $95.88 \pm 0.20$ | $66.92 \pm 1.18$ |
|  | NatPN | $90.04 \pm 0.38$ | $75.31 \pm 1.00$ | $96.79 \pm 0.22$ | $47.23 \pm 2.53$ |
|  | GPN | $88.94 \pm 0.64$ | $67.40 \pm 0.86$ | $95.57 \pm 0.26$ | $90.45 \pm 1.76$ |
|  | NatGPN | $80.45 \pm 6.19$ | $70.93 \pm 1.25$ | $91.40 \pm 3.66$ | $82.22 \pm 4.95$ |
|  | EnsGNN | $91.35 \pm n/a$ | $74.81 \pm n/a$ | $97.26 \pm n/a$ | $65.54 \pm n/a$ |
|  | EnsNatPN | $90.75 \pm n/a$ | $74.69 \pm n/a$ | $97.02 \pm n/a$ | $42.47 \pm n/a$ |
|  | EnsGPN | $89.75 \pm n/a$ | $66.27 \pm n/a$ | $95.85 \pm n/a$ | $92.39 \pm n/a$ |
| **PPR** | MLP | $49.68 \pm 0.80$ | $45.47 \pm 3.75$ | $61.05 \pm 0.60$ | $78.29 \pm 1.27$ |
|  | GNN | $53.65 \pm 2.91$ | $54.98 \pm 10.32$ | $67.28 \pm 3.57$ | $86.94 \pm 1.93$ |
|  | EN | $52.89 \pm 3.08$ | $47.70 \pm 6.44$ | $64.74 \pm 3.15$ | $88.03 \pm 1.88$ |
|  | PN | $60.22 \pm 2.81$ | $50.42 \pm 11.67$ | $72.29 \pm 3.33$ | $82.47 \pm 2.45$ |
|  | NatPN | $53.15 \pm 1.93$ | $62.85 \pm 4.67$ | $68.78 \pm 1.24$ | $49.36 \pm 11.49$ |
|  | GPN | $70.16 \pm 0.36$ | $33.12 \pm 16.84$ | $77.10 \pm 3.51$ | $91.90 \pm 3.24$ |
|  | NatGPN | $48.62 \pm 2.94$ | $77.97 \pm 6.97$ | $68.04 \pm 2.65$ | $91.17 \pm 1.60$ |
|  | EnsGNN | $54.56 \pm n/a$ | $58.28 \pm n/a$ | $69.01 \pm n/a$ | $89.05 \pm n/a$ |
|  | EnsNatPN | $55.24 \pm n/a$ | $62.02 \pm n/a$ | $70.58 \pm n/a$ | $43.71 \pm n/a$ |
|  | EnsGPN | $70.30 \pm n/a$ | $34.13 \pm n/a$ | $77.43 \pm n/a$ | $92.21 \pm n/a$ |

Table 18: Experiment results on **CoauthorCS** dataset.

|  |  | Accuracy | PRR@TU | AUPRC@TU | AUROC@KU |
|---|---|---|---|---|---|
| **Random** | MLP | $93.28 \pm 0.11$ | $64.47 \pm 1.27$ | $97.32 \pm 0.05$ | $n/a$ |
| | GNN | $93.41 \pm 0.08$ | $73.37 \pm 1.09$ | $97.93 \pm 0.05$ | $n/a$ |
| | EN | $93.33 \pm 0.12$ | $74.24 \pm 0.84$ | $97.95 \pm 0.05$ | $n/a$ |
| | PN | $90.29 \pm 0.78$ | $75.42 \pm 0.98$ | $96.90 \pm 0.29$ | $n/a$ |
| | NatPN | $93.42 \pm 0.08$ | $77.37 \pm 0.34$ | $98.18 \pm 0.04$ | $n/a$ |
| | GPN | $90.51 \pm 0.15$ | $65.11 \pm 0.84$ | $96.10 \pm 0.10$ | $n/a$ |
| | NatGPN | $88.91 \pm 0.28$ | $63.57 \pm 1.44$ | $95.18 \pm 0.24$ | $n/a$ |
| | EnsGNN | $93.61 \pm n/a$ | $73.36 \pm n/a$ | $98.00 \pm n/a$ | $n/a$ |
| | EnsNatPN | $93.50 \pm n/a$ | $77.84 \pm n/a$ | $98.23 \pm n/a$ | $n/a$ |
| | EnsGPN | $90.89 \pm n/a$ | $64.94 \pm n/a$ | $96.27 \pm n/a$ | $n/a$ |
| **Feature** | MLP | $92.42 \pm 0.20$ | $62.88 \pm 1.39$ | $96.83 \pm 0.08$ | $64.00 \pm 0.40$ |
| | GNN | $93.48 \pm 0.11$ | $71.83 \pm 0.70$ | $97.86 \pm 0.04$ | $50.67 \pm 0.82$ |
| | EN | $93.44 \pm 0.07$ | $72.68 \pm 0.63$ | $97.90 \pm 0.02$ | $51.47 \pm 0.95$ |
| | PN | $89.72 \pm 0.52$ | $76.57 \pm 1.28$ | $96.78 \pm 0.24$ | $50.25 \pm 0.68$ |
| | NatPN | $93.60 \pm 0.09$ | $75.68 \pm 0.31$ | $98.13 \pm 0.03$ | $44.86 \pm 1.22$ |
| | GPN | $90.65 \pm 0.18$ | $66.68 \pm 0.39$ | $96.30 \pm 0.08$ | $60.59 \pm 0.48$ |
| | NatGPN | $88.07 \pm 0.43$ | $67.63 \pm 2.67$ | $95.18 \pm 0.20$ | $61.80 \pm 0.87$ |
| | EnsGNN | $93.72 \pm n/a$ | $71.61 \pm n/a$ | $97.93 \pm n/a$ | $49.91 \pm n/a$ |
| | EnsNatPN | $93.64 \pm n/a$ | $76.41 \pm n/a$ | $98.19 \pm n/a$ | $43.61 \pm n/a$ |
| | EnsGPN | $91.00 \pm n/a$ | $67.16 \pm n/a$ | $96.50 \pm n/a$ | $60.47 \pm n/a$ |
| **PageRank** | MLP | $93.15 \pm 0.25$ | $60.50 \pm 1.14$ | $97.01 \pm 0.08$ | $68.00 \pm 0.38$ |
| | GNN | $92.03 \pm 0.12$ | $70.03 \pm 0.58$ | $97.17 \pm 0.06$ | $52.17 \pm 0.29$ |
| | EN | $92.01 \pm 0.15$ | $70.48 \pm 0.83$ | $97.19 \pm 0.05$ | $52.92 \pm 0.50$ |
| | PN | $88.46 \pm 0.66$ | $74.08 \pm 1.62$ | $96.02 \pm 0.29$ | $60.17 \pm 0.88$ |
| | NatPN | $92.12 \pm 0.05$ | $74.94 \pm 0.37$ | $97.56 \pm 0.03$ | $48.79 \pm 1.89$ |
| | GPN | $88.82 \pm 0.40$ | $67.52 \pm 0.64$ | $95.53 \pm 0.18$ | $82.05 \pm 2.57$ |
| | NatGPN | $86.51 \pm 0.37$ | $61.35 \pm 1.43$ | $93.67 \pm 0.30$ | $79.99 \pm 1.09$ |
| | EnsGNN | $92.05 \pm n/a$ | $70.57 \pm n/a$ | $97.21 \pm n/a$ | $51.54 \pm n/a$ |
| | EnsNatPN | $92.22 \pm n/a$ | $75.38 \pm n/a$ | $97.63 \pm n/a$ | $48.16 \pm n/a$ |
| | EnsGPN | $89.12 \pm n/a$ | $68.56 \pm n/a$ | $95.77 \pm n/a$ | $80.38 \pm n/a$ |
| **PPR** | MLP | $90.46 \pm 0.35$ | $61.93 \pm 1.70$ | $95.81 \pm 0.10$ | $73.96 \pm 1.06$ |
| | GNN | $91.84 \pm 0.13$ | $67.13 \pm 0.82$ | $96.87 \pm 0.08$ | $66.72 \pm 1.59$ |
| | EN | $91.77 \pm 0.09$ | $67.66 \pm 0.81$ | $96.88 \pm 0.08$ | $67.42 \pm 2.19$ |
| | PN | $85.21 \pm 0.79$ | $73.32 \pm 1.56$ | $94.44 \pm 0.56$ | $66.03 \pm 2.59$ |
| | NatPN | $92.29 \pm 0.08$ | $74.19 \pm 0.79$ | $97.57 \pm 0.08$ | $42.36 \pm 2.24$ |
| | GPN | $88.31 \pm 0.38$ | $71.15 \pm 0.29$ | $95.65 \pm 0.18$ | $88.38 \pm 0.78$ |
| | NatGPN | $85.07 \pm 0.48$ | $63.26 \pm 2.25$ | $93.10 \pm 0.39$ | $80.75 \pm 1.98$ |
| | EnsGNN | $92.05 \pm n/a$ | $66.82 \pm n/a$ | $96.94 \pm n/a$ | $71.42 \pm n/a$ |
| | EnsNatPN | $92.41 \pm n/a$ | $74.70 \pm n/a$ | $97.65 \pm n/a$ | $40.08 \pm n/a$ |
| | EnsGPN | $88.74 \pm n/a$ | $72.05 \pm n/a$ | $95.94 \pm n/a$ | $88.81 \pm n/a$ |

Table 19: Experiment results on **CoauthorPhysics** dataset.

| | | Accuracy | PRR@TU | AUPRC@TU | AUROC@KU |
|---|---|---|---|---|---|
| **Random** | MLP | $95.53 \pm 0.03$ | $75.21 \pm 0.34$ | $98.74 \pm 0.02$ | $n/a$ |
| | GNN | $96.32 \pm 0.06$ | $81.17 \pm 0.47$ | $99.20 \pm 0.01$ | $n/a$ |
| | EN | $96.35 \pm 0.04$ | $81.01 \pm 0.92$ | $99.20 \pm 0.03$ | $n/a$ |
| | PN | $95.28 \pm 0.26$ | $80.14 \pm 1.01$ | $98.88 \pm 0.10$ | $n/a$ |
| | NatPN | $96.31 \pm 0.05$ | $82.37 \pm 0.61$ | $99.24 \pm 0.02$ | $n/a$ |
| | GPN | $95.32 \pm 0.08$ | $79.13 \pm 0.44$ | $98.85 \pm 0.03$ | $n/a$ |
| | NatGPN | $94.37 \pm 0.32$ | $79.53 \pm 1.17$ | $98.59 \pm 0.16$ | $n/a$ |
| | EnsGNN | $96.33 \pm n/a$ | $81.47 \pm n/a$ | $99.21 \pm n/a$ | $n/a$ |
| | EnsNatPN | $96.30 \pm n/a$ | $82.77 \pm n/a$ | $99.25 \pm n/a$ | $n/a$ |
| | EnsGPN | $95.45 \pm n/a$ | $79.41 \pm n/a$ | $98.90 \pm n/a$ | $n/a$ |
| **Feature** | MLP | $95.54 \pm 0.03$ | $75.55 \pm 0.27$ | $98.76 \pm 0.02$ | $52.04 \pm 0.08$ |
| | GNN | $96.16 \pm 0.05$ | $81.91 \pm 0.73$ | $99.18 \pm 0.03$ | $51.82 \pm 0.25$ |
| | EN | $96.13 \pm 0.05$ | $82.17 \pm 1.06$ | $99.19 \pm 0.04$ | $51.73 \pm 0.46$ |
| | PN | $95.31 \pm 0.16$ | $80.88 \pm 1.07$ | $98.93 \pm 0.08$ | $51.48 \pm 0.57$ |
| | NatPN | $96.09 \pm 0.06$ | $84.92 \pm 0.27$ | $99.28 \pm 0.02$ | $48.32 \pm 0.54$ |
| | GPN | $95.12 \pm 0.07$ | $82.13 \pm 0.57$ | $98.93 \pm 0.02$ | $54.36 \pm 0.22$ |
| | NatGPN | $93.92 \pm 0.36$ | $80.34 \pm 0.79$ | $98.50 \pm 0.15$ | $54.30 \pm 0.31$ |
| | EnsGNN | $96.16 \pm n/a$ | $82.27 \pm n/a$ | $99.20 \pm n/a$ | $51.11 \pm n/a$ |
| | EnsNatPN | $96.10 \pm n/a$ | $85.17 \pm n/a$ | $99.29 \pm n/a$ | $47.89 \pm n/a$ |
| | EnsGPN | $95.15 \pm n/a$ | $82.75 \pm n/a$ | $98.97 \pm n/a$ | $54.50 \pm n/a$ |
| **PageRank** | MLP | $94.53 \pm 0.09$ | $72.39 \pm 0.40$ | $98.27 \pm 0.03$ | $73.99 \pm 0.28$ |
| | GNN | $95.06 \pm 0.07$ | $76.52 \pm 0.27$ | $98.65 \pm 0.02$ | $60.23 \pm 0.20$ |
| | EN | $95.07 \pm 0.05$ | $76.70 \pm 0.82$ | $98.66 \pm 0.03$ | $61.02 \pm 1.03$ |
| | PN | $93.43 \pm 0.39$ | $76.39 \pm 1.30$ | $98.12 \pm 0.16$ | $65.12 \pm 1.62$ |
| | NatPN | $95.08 \pm 0.06$ | $78.15 \pm 0.69$ | $98.73 \pm 0.03$ | $47.06 \pm 3.39$ |
| | GPN | $93.41 \pm 0.08$ | $75.97 \pm 0.33$ | $98.09 \pm 0.04$ | $78.31 \pm 0.88$ |
| | NatGPN | $91.68 \pm 0.58$ | $72.12 \pm 1.44$ | $97.17 \pm 0.35$ | $82.95 \pm 3.02$ |
| | EnsGNN | $95.11 \pm n/a$ | $76.61 \pm n/a$ | $98.67 \pm n/a$ | $58.45 \pm n/a$ |
| | EnsNatPN | $95.11 \pm n/a$ | $78.39 \pm n/a$ | $98.75 \pm n/a$ | $45.49 \pm n/a$ |
| | EnsGPN | $93.56 \pm n/a$ | $76.09 \pm n/a$ | $98.14 \pm n/a$ | $78.72 \pm n/a$ |
| **PPR** | MLP | $89.11 \pm 0.24$ | $45.53 \pm 2.19$ | $93.53 \pm 0.29$ | $91.39 \pm 0.19$ |
| | GNN | $94.49 \pm 0.15$ | $57.61 \pm 2.31$ | $97.49 \pm 0.13$ | $90.71 \pm 0.12$ |
| | EN | $94.26 \pm 0.34$ | $54.63 \pm 2.40$ | $97.22 \pm 0.21$ | $90.69 \pm 0.34$ |
| | PN | $88.58 \pm 1.89$ | $59.44 \pm 4.95$ | $94.60 \pm 0.92$ | $81.37 \pm 4.48$ |
| | NatPN | $94.53 \pm 0.10$ | $68.02 \pm 0.75$ | $98.05 \pm 0.07$ | $34.41 \pm 6.64$ |
| | GPN | $90.31 \pm 0.71$ | $67.05 \pm 0.76$ | $96.17 \pm 0.37$ | $91.95 \pm 1.21$ |
| | NatGPN | $88.29 \pm 1.37$ | $49.35 \pm 2.15$ | $93.39 \pm 0.90$ | $86.99 \pm 1.45$ |
| | EnsGNN | $94.58 \pm n/a$ | $57.81 \pm n/a$ | $97.54 \pm n/a$ | $90.42 \pm n/a$ |
| | EnsNatPN | $94.67 \pm n/a$ | $67.86 \pm n/a$ | $98.09 \pm n/a$ | $27.08 \pm n/a$ |
| | EnsGPN | $90.63 \pm n/a$ | $68.17 \pm n/a$ | $96.42 \pm n/a$ | $92.58 \pm n/a$ |

Table 20: Experiment results on **CoraML** dataset.

| | | Accuracy | PRR@TU | AUPRC@TU | AUROC@KU |
|---|---|---|---|---|---|
| **Random** | MLP | $72.08 \pm 0.65$ | $57.70 \pm 1.27$ | $83.69 \pm 0.41$ | $n/a$ |
| | GNN | $86.95 \pm 0.33$ | $70.63 \pm 1.64$ | $94.97 \pm 0.12$ | $n/a$ |
| | EN | $87.01 \pm 0.33$ | $70.14 \pm 2.07$ | $94.94 \pm 0.13$ | $n/a$ |
| | PN | $78.81 \pm 0.79$ | $62.59 \pm 3.10$ | $89.26 \pm 0.71$ | $n/a$ |
| | NatPN | $87.08 \pm 0.66$ | $70.83 \pm 2.21$ | $95.06 \pm 0.12$ | $n/a$ |
| | GPN | $85.42 \pm 0.17$ | $65.41 \pm 0.94$ | $93.57 \pm 0.08$ | $n/a$ |
| | NatGPN | $86.28 \pm 0.18$ | $63.90 \pm 1.44$ | $93.84 \pm 0.17$ | $n/a$ |
| | EnsGNN | $87.31 \pm n/a$ | $71.37 \pm n/a$ | $95.22 \pm n/a$ | $n/a$ |
| | EnsNatPN | $87.44 \pm n/a$ | $71.43 \pm n/a$ | $95.29 \pm n/a$ | $n/a$ |
| | EnsGPN | $85.64 \pm n/a$ | $64.25 \pm n/a$ | $93.54 \pm n/a$ | $n/a$ |
| **Feature** | MLP | $71.66 \pm 1.08$ | $54.74 \pm 1.24$ | $82.78 \pm 0.60$ | $62.42 \pm 0.49$ |
| | GNN | $86.31 \pm 0.44$ | $66.29 \pm 2.42$ | $94.15 \pm 0.11$ | $52.56 \pm 0.55$ |
| | EN | $86.25 \pm 0.33$ | $66.77 \pm 1.59$ | $94.17 \pm 0.10$ | $52.70 \pm 0.38$ |
| | PN | $77.61 \pm 1.77$ | $62.83 \pm 3.97$ | $88.52 \pm 1.21$ | $49.55 \pm 0.67$ |
| | NatPN | $86.09 \pm 0.32$ | $67.00 \pm 2.20$ | $94.12 \pm 0.24$ | $43.51 \pm 1.20$ |
| | GPN | $84.58 \pm 0.22$ | $61.94 \pm 1.00$ | $92.66 \pm 0.17$ | $54.97 \pm 0.88$ |
| | NatGPN | $85.16 \pm 0.40$ | $63.19 \pm 1.60$ | $93.14 \pm 0.24$ | $55.16 \pm 0.80$ |
| | EnsGNN | $86.91 \pm n/a$ | $66.17 \pm n/a$ | $94.44 \pm n/a$ | $50.77 \pm n/a$ |
| | EnsNatPN | $86.57 \pm n/a$ | $66.70 \pm n/a$ | $94.33 \pm n/a$ | $40.67 \pm n/a$ |
| | EnsGPN | $84.30 \pm n/a$ | $62.47 \pm n/a$ | $92.57 \pm n/a$ | $54.24 \pm n/a$ |
| **PageRank** | MLP | $70.14 \pm 0.66$ | $56.09 \pm 1.64$ | $81.89 \pm 0.23$ | $56.16 \pm 0.61$ |
| | GNN | $84.29 \pm 0.14$ | $66.28 \pm 1.00$ | $93.07 \pm 0.09$ | $53.23 \pm 0.65$ |
| | EN | $84.57 \pm 0.36$ | $65.17 \pm 1.12$ | $93.07 \pm 0.07$ | $53.27 \pm 0.59$ |
| | PN | $74.96 \pm 1.50$ | $56.41 \pm 3.92$ | $85.53 \pm 1.57$ | $59.91 \pm 1.52$ |
| | NatPN | $84.82 \pm 0.30$ | $64.28 \pm 1.05$ | $93.10 \pm 0.10$ | $59.95 \pm 2.45$ |
| | GPN | $83.87 \pm 0.28$ | $65.16 \pm 1.11$ | $92.69 \pm 0.24$ | $82.48 \pm 2.87$ |
| | NatGPN | $83.74 \pm 0.28$ | $66.33 \pm 1.04$ | $92.77 \pm 0.22$ | $74.32 \pm 1.97$ |
| | EnsGNN | $84.70 \pm n/a$ | $67.09 \pm n/a$ | $93.40 \pm n/a$ | $54.17 \pm n/a$ |
| | EnsNatPN | $84.97 \pm n/a$ | $65.42 \pm n/a$ | $93.32 \pm n/a$ | $65.41 \pm n/a$ |
| | EnsGPN | $83.77 \pm n/a$ | $64.28 \pm n/a$ | $92.51 \pm n/a$ | $81.42 \pm n/a$ |
| **PPR** | MLP | $56.41 \pm 0.41$ | $37.24 \pm 0.92$ | $65.56 \pm 0.47$ | $63.84 \pm 0.53$ |
| | GNN | $77.76 \pm 0.60$ | $58.02 \pm 3.22$ | $87.79 \pm 0.50$ | $69.36 \pm 0.82$ |
| | EN | $77.33 \pm 0.72$ | $59.21 \pm 2.36$ | $87.71 \pm 0.35$ | $68.21 \pm 0.94$ |
| | PN | $59.39 \pm 2.60$ | $51.36 \pm 6.21$ | $71.73 \pm 3.51$ | $72.71 \pm 4.29$ |
| | NatPN | $78.40 \pm 0.74$ | $60.01 \pm 1.57$ | $88.56 \pm 0.34$ | $58.37 \pm 4.00$ |
| | GPN | $77.54 \pm 1.46$ | $59.74 \pm 1.79$ | $87.93 \pm 1.04$ | $92.77 \pm 0.77$ |
| | NatGPN | $77.49 \pm 1.37$ | $56.85 \pm 3.25$ | $87.39 \pm 1.24$ | $84.86 \pm 1.65$ |
| | EnsGNN | $79.16 \pm n/a$ | $56.34 \pm n/a$ | $88.45 \pm n/a$ | $73.52 \pm n/a$ |
| | EnsNatPN | $79.23 \pm n/a$ | $59.96 \pm n/a$ | $89.09 \pm n/a$ | $63.61 \pm n/a$ |
| | EnsGPN | $78.62 \pm n/a$ | $60.95 \pm n/a$ | $88.87 \pm n/a$ | $92.32 \pm n/a$ |

Table 21: Experiment results on **CiteSeer** dataset.

|  |  | Accuracy | PRR@TU | AUPRC@TU | AUROC@KU |
|---|---|---|---|---|---|
| **Random** | MLP | $66.57 \pm 0.33$ | $53.58 \pm 0.56$ | $78.50 \pm 0.27$ | $n/a$ |
|  | GNN | $72.39 \pm 0.44$ | $56.19 \pm 1.65$ | $83.62 \pm 0.16$ | $n/a$ |
|  | EN | $72.34 \pm 0.53$ | $56.17 \pm 1.67$ | $83.58 \pm 0.20$ | $n/a$ |
|  | PN | $64.44 \pm 0.86$ | $48.78 \pm 1.91$ | $75.61 \pm 0.86$ | $n/a$ |
|  | NatPN | $72.52 \pm 0.49$ | $55.19 \pm 1.83$ | $83.52 \pm 0.41$ | $n/a$ |
|  | GPN | $69.94 \pm 0.42$ | $50.41 \pm 1.42$ | $80.54 \pm 0.22$ | $n/a$ |
|  | NatGPN | $70.88 \pm 0.58$ | $50.44 \pm 0.64$ | $81.29 \pm 0.56$ | $n/a$ |
|  | EnsGNN | $73.50 \pm n/a$ | $55.15 \pm n/a$ | $84.24 \pm n/a$ | $n/a$ |
|  | EnsNatPN | $73.20 \pm n/a$ | $53.89 \pm n/a$ | $83.77 \pm n/a$ | $n/a$ |
|  | EnsGPN | $70.43 \pm n/a$ | $49.69 \pm n/a$ | $80.78 \pm n/a$ | $n/a$ |
| **Feature** | MLP | $64.70 \pm 0.36$ | $50.93 \pm 1.11$ | $76.33 \pm 0.17$ | $55.84 \pm 0.19$ |
|  | GNN | $71.37 \pm 0.23$ | $49.62 \pm 1.14$ | $81.51 \pm 0.26$ | $51.09 \pm 0.91$ |
|  | EN | $71.26 \pm 0.39$ | $49.82 \pm 1.60$ | $81.46 \pm 0.29$ | $50.96 \pm 1.27$ |
|  | PN | $62.21 \pm 0.49$ | $48.65 \pm 2.11$ | $73.65 \pm 0.48$ | $50.70 \pm 0.24$ |
|  | NatPN | $71.50 \pm 0.56$ | $49.10 \pm 0.67$ | $81.51 \pm 0.35$ | $47.09 \pm 1.81$ |
|  | GPN | $68.00 \pm 0.46$ | $42.05 \pm 1.82$ | $77.15 \pm 0.17$ | $51.76 \pm 0.37$ |
|  | NatGPN | $69.15 \pm 0.32$ | $44.22 \pm 1.07$ | $78.58 \pm 0.16$ | $50.30 \pm 1.40$ |
|  | EnsGNN | $72.42 \pm n/a$ | $47.84 \pm n/a$ | $81.97 \pm n/a$ | $50.18 \pm n/a$ |
|  | EnsNatPN | $72.12 \pm n/a$ | $48.70 \pm n/a$ | $81.91 \pm n/a$ | $45.65 \pm n/a$ |
|  | EnsGPN | $68.27 \pm n/a$ | $42.19 \pm n/a$ | $77.41 \pm n/a$ | $51.62 \pm n/a$ |
| **PageRank** | MLP | $65.96 \pm 0.34$ | $46.36 \pm 0.54$ | $76.37 \pm 0.22$ | $54.33 \pm 0.23$ |
|  | GNN | $72.03 \pm 0.31$ | $51.64 \pm 1.39$ | $82.44 \pm 0.19$ | $50.82 \pm 0.43$ |
|  | EN | $71.79 \pm 0.23$ | $52.28 \pm 1.41$ | $82.38 \pm 0.16$ | $50.73 \pm 0.78$ |
|  | PN | $62.51 \pm 0.98$ | $46.12 \pm 2.34$ | $73.32 \pm 0.65$ | $56.33 \pm 1.41$ |
|  | NatPN | $72.62 \pm 0.10$ | $50.76 \pm 1.02$ | $82.71 \pm 0.17$ | $54.78 \pm 1.55$ |
|  | GPN | $70.84 \pm 0.46$ | $41.23 \pm 2.81$ | $79.36 \pm 0.26$ | $74.78 \pm 1.42$ |
|  | NatGPN | $71.65 \pm 0.39$ | $42.54 \pm 1.84$ | $80.29 \pm 0.34$ | $59.46 \pm 3.89$ |
|  | EnsGNN | $72.48 \pm n/a$ | $52.17 \pm n/a$ | $82.88 \pm n/a$ | $52.31 \pm n/a$ |
|  | EnsNatPN | $72.90 \pm n/a$ | $51.35 \pm n/a$ | $83.04 \pm n/a$ | $57.63 \pm n/a$ |
|  | EnsGPN | $70.91 \pm n/a$ | $42.07 \pm n/a$ | $79.59 \pm n/a$ | $77.27 \pm n/a$ |
| **PPR** | MLP | $55.53 \pm 0.28$ | $39.66 \pm 1.44$ | $65.32 \pm 0.23$ | $69.37 \pm 0.39$ |
|  | GNN | $62.22 \pm 0.60$ | $44.67 \pm 2.44$ | $72.72 \pm 0.68$ | $81.83 \pm 0.54$ |
|  | EN | $62.18 \pm 0.91$ | $44.78 \pm 1.08$ | $72.70 \pm 0.89$ | $80.35 \pm 0.77$ |
|  | PN | $46.30 \pm 2.63$ | $33.99 \pm 8.13$ | $54.74 \pm 4.41$ | $80.83 \pm 1.27$ |
|  | NatPN | $63.09 \pm 0.58$ | $41.88 \pm 1.41$ | $72.84 \pm 0.37$ | $47.42 \pm 6.69$ |
|  | GPN | $56.27 \pm 1.23$ | $40.11 \pm 3.87$ | $66.14 \pm 0.75$ | $93.34 \pm 4.34$ |
|  | NatGPN | $64.00 \pm 1.40$ | $36.33 \pm 0.48$ | $72.37 \pm 1.17$ | $88.78 \pm 4.75$ |
|  | EnsGNN | $64.48 \pm n/a$ | $43.41 \pm n/a$ | $74.42 \pm n/a$ | $85.18 \pm n/a$ |
|  | EnsNatPN | $64.54 \pm n/a$ | $42.71 \pm n/a$ | $74.32 \pm n/a$ | $44.31 \pm n/a$ |
|  | EnsGPN | $59.38 \pm n/a$ | $37.66 \pm n/a$ | $68.46 \pm n/a$ | $92.86 \pm n/a$ |

Table 22: Experiment results on **PubMed** dataset.

| | | Accuracy | PRR@TU | AUPRC@TU | AUROC@KU |
|---|---|---|---|---|---|
| **Random** | MLP | $85.59 \pm 0.20$ | $58.82 \pm 0.43$ | $92.85 \pm 0.06$ | $n/a$ |
| | GNN | $86.74 \pm 0.22$ | $64.45 \pm 0.74$ | $94.15 \pm 0.12$ | $n/a$ |
| | EN | $86.90 \pm 0.28$ | $64.78 \pm 0.77$ | $94.27 \pm 0.12$ | $n/a$ |
| | PN | $85.89 \pm 0.38$ | $57.80 \pm 1.45$ | $92.89 \pm 0.17$ | $n/a$ |
| | NatPN | $86.58 \pm 0.31$ | $63.84 \pm 0.84$ | $94.00 \pm 0.11$ | $n/a$ |
| | GPN | $87.34 \pm 0.23$ | $63.46 \pm 0.41$ | $94.36 \pm 0.11$ | $n/a$ |
| | NatGPN | $85.08 \pm 0.81$ | $60.22 \pm 1.96$ | $92.71 \pm 0.72$ | $n/a$ |
| | EnsGNN | $87.43 \pm n/a$ | $66.12 \pm n/a$ | $94.70 \pm n/a$ | $n/a$ |
| | EnsNatPN | $87.15 \pm n/a$ | $65.21 \pm n/a$ | $94.45 \pm n/a$ | $n/a$ |
| | EnsGPN | $87.39 \pm n/a$ | $64.30 \pm n/a$ | $94.48 \pm n/a$ | $n/a$ |
| **Feature** | MLP | $84.66 \pm 0.14$ | $60.50 \pm 0.34$ | $92.52 \pm 0.04$ | $48.63 \pm 0.09$ |
| | GNN | $86.32 \pm 0.17$ | $65.27 \pm 0.39$ | $94.03 \pm 0.06$ | $50.50 \pm 0.24$ |
| | EN | $86.36 \pm 0.30$ | $65.81 \pm 1.03$ | $94.11 \pm 0.06$ | $50.27 \pm 0.31$ |
| | PN | $85.46 \pm 0.19$ | $58.47 \pm 1.87$ | $92.73 \pm 0.24$ | $49.03 \pm 0.84$ |
| | NatPN | $86.15 \pm 0.17$ | $64.99 \pm 0.40$ | $93.91 \pm 0.06$ | $50.73 \pm 0.28$ |
| | GPN | $86.89 \pm 0.29$ | $62.46 \pm 0.73$ | $94.00 \pm 0.14$ | $52.47 \pm 0.37$ |
| | NatGPN | $84.76 \pm 1.06$ | $60.53 \pm 1.60$ | $92.56 \pm 0.84$ | $50.83 \pm 0.28$ |
| | EnsGNN | $87.15 \pm n/a$ | $66.24 \pm n/a$ | $94.57 \pm n/a$ | $50.33 \pm n/a$ |
| | EnsNatPN | $86.67 \pm n/a$ | $66.59 \pm n/a$ | $94.36 \pm n/a$ | $51.47 \pm n/a$ |
| | EnsGPN | $87.17 \pm n/a$ | $62.69 \pm n/a$ | $94.18 \pm n/a$ | $52.77 \pm n/a$ |
| **PageRank** | MLP | $85.31 \pm 0.11$ | $58.78 \pm 0.43$ | $92.68 \pm 0.05$ | $51.32 \pm 0.09$ |
| | GNN | $84.45 \pm 0.20$ | $60.85 \pm 0.86$ | $92.44 \pm 0.19$ | $55.99 \pm 0.39$ |
| | EN | $84.77 \pm 0.33$ | $60.97 \pm 0.57$ | $92.64 \pm 0.21$ | $56.80 \pm 0.38$ |
| | PN | $83.12 \pm 0.25$ | $52.37 \pm 1.29$ | $90.47 \pm 0.23$ | $66.37 \pm 3.42$ |
| | NatPN | $84.29 \pm 0.38$ | $61.00 \pm 0.40$ | $92.37 \pm 0.20$ | $56.87 \pm 1.97$ |
| | GPN | $86.06 \pm 0.18$ | $63.96 \pm 0.42$ | $93.73 \pm 0.12$ | $82.67 \pm 0.59$ |
| | NatGPN | $82.66 \pm 0.79$ | $59.50 \pm 2.85$ | $91.17 \pm 0.91$ | $74.55 \pm 1.56$ |
| | EnsGNN | $85.09 \pm n/a$ | $63.43 \pm n/a$ | $93.14 \pm n/a$ | $60.77 \pm n/a$ |
| | EnsNatPN | $84.88 \pm n/a$ | $62.66 \pm n/a$ | $92.92 \pm n/a$ | $61.38 \pm n/a$ |
| | EnsGPN | $86.41 \pm n/a$ | $64.25 \pm n/a$ | $93.95 \pm n/a$ | $83.77 \pm n/a$ |
| **PPR** | MLP | $84.12 \pm 0.05$ | $53.29 \pm 0.34$ | $91.24 \pm 0.05$ | $56.44 \pm 0.24$ |
| | GNN | $84.80 \pm 0.20$ | $60.82 \pm 0.70$ | $92.64 \pm 0.10$ | $58.41 \pm 1.03$ |
| | EN | $85.04 \pm 0.31$ | $61.21 \pm 0.65$ | $92.83 \pm 0.16$ | $66.78 \pm 2.97$ |
| | PN | $83.31 \pm 0.40$ | $48.27 \pm 1.77$ | $90.02 \pm 0.47$ | $72.08 \pm 1.95$ |
| | NatPN | $84.50 \pm 0.43$ | $60.66 \pm 1.06$ | $92.44 \pm 0.24$ | $61.40 \pm 4.36$ |
| | GPN | $86.05 \pm 0.14$ | $62.12 \pm 1.35$ | $93.50 \pm 0.13$ | $74.88 \pm 1.93$ |
| | NatGPN | $82.85 \pm 1.88$ | $57.71 \pm 2.81$ | $91.01 \pm 1.60$ | $68.86 \pm 0.67$ |
| | EnsGNN | $85.82 \pm n/a$ | $62.38 \pm n/a$ | $93.41 \pm n/a$ | $62.02 \pm n/a$ |
| | EnsNatPN | $85.18 \pm n/a$ | $62.19 \pm n/a$ | $93.03 \pm n/a$ | $64.73 \pm n/a$ |
| | EnsGPN | $86.18 \pm n/a$ | $62.76 \pm n/a$ | $93.65 \pm n/a$ | $74.62 \pm n/a$ |

# E  COMPONENTS OF DIRICHLET-BASED FRAMEWORK

## E.1  STANDARD OR NATURAL DENSITY ESTIMATION

One property that we vary in these methods is the number of Normalizing Flows and how we use their density estimates. In particular, we consider a *Standard* approach (Charpentier et al., 2020), where distinct Normalizing Flows $\mathrm{p}_\psi(z_i|k)$ are used per each class $k$ to predict the corresponding Dirichlet parameter $\beta_{ik}^{\mathrm{feat}}$ based on the representation $z_i = f_\phi(x_i)$:

$$\beta_{ik}^{\mathrm{feat}} = n \cdot \mathrm{p}_\psi(z_i, k) = n \cdot \mathrm{p}_\psi(z_i|k) \cdot \mathrm{P}(k) = n_k \cdot \mathrm{p}_\psi(z_i|k),$$

where the probability $\mathrm{P}(k)$ of class $k$ is approximated by the ratio of train observations from class $k$, $n$ is usually equal to the dataset size (but can, in general, be a hyper-parameter), and $n_k := n \cdot \mathrm{P}(k)$.

Another approach is the *Natural* version of Posterior Network that is proposed by Charpentier et al. (2021). It exploits a single Normalizing Flow $\mathrm{p}_\psi(z_i)$ to predict the evidence $S_i$, while the normalized Dirichlet parameters $\mu_i$ are obtained using one-layer linear transformation $g_\omega(z_i)$. The final predictions are obtained as follows:

$$S_i = n \cdot \mathrm{p}_\psi(z_i), \ \mu_i = g_\omega(z_i) \Longrightarrow \beta_{ik}^{\mathrm{feat}} = S_i \cdot \mu_{ik}$$

## E.2  GRAPH ENCODING OR GRAPH PROPAGATION

Another aspect that we analyze is how the graph structure can be combined with Posterior Networks. Here, we consider two approaches — use the graph for the *encoding* to obtain $z_i$ or use *graph propagation* to smooth the predicted parameters as the post-processing step.

In the case of graph encoding, we use a two-layer SAGE convolution to produce the representations $z_i$ and then estimate the density as usual via Normalizing Flows. Thus, the graph structure is used for the pre-processing step.

The second approach is adopted by Stadler et al. (2021). In this case, the initial representations $z_i$ are obtained using a graph-agnostic two-layer MLP, while graph propagation is applied to smooth the Dirichlet parameters $\beta_i^{\mathrm{feat}}$. Graph propagation is performed in several steps via some transformation $\pi : \mathbb{R}^{n \times C} \to \mathbb{R}^{n \times C}$ as follows:

$$\mathbf{B}^{t+1} = (1 - \alpha)\pi(\mathbf{B}^t) + \alpha\mathbf{B},$$

where $\mathbf{B}^0 = \mathbf{B} \in \mathbb{R}^{n \times C}$ is formed by the parameters $\beta_{ik}^{\mathrm{feat}}$ and $\alpha$ is a hyperparameter that controls the smoothing effect of the propagation step. Similarly to Stadler et al. (2021), we use a *Personalized Propagation* scheme (Klicpera et al., 2018), which takes into account the mutual importance of nodes and is defined as $\pi(\mathbf{B}) = \mathbf{D}^{-1/2}\mathbf{A}\mathbf{D}^{-1/2}\mathbf{B}$.

We refer to the method with graph propagation as **GPN** (Graph Posterior Network) and with Graph Encoding as **PN** (Posterior Network). Further, if the Natural version of the Posterior Network is used instead of the Standard one, we denote the models **NatGPN** and **NatPN**, respectively.

# F  DATASET DETAILS

Table 23: Description of the considered graph datasets for node classification task.

|  | #Nodes | #Edges | #Classes | #Features |
|---|---|---|---|---|
| AmazonComputers | 13,381 | 259,159 | 10 | 767 |
| AmazonPhoto | 7,484 | 126,530 | 8 | 745 |
| CoauthorCS | 18,333 | 163,788 | 15 | 6,805 |
| CoauthorPhysics | 34,493 | 495,924 | 5 | 8,415 |
| CoraML | 2,995 | 16,316 | 7 | 2,879 |
| CiteSeer | 3,327 | 4,732 | 6 | 3,703 |
| PubMed | 19,717 | 44,338 | 3 | 8,415 |

## G    EXPERIMENTAL SETUP

We evaluate all the methods discussed in Section 4 using the benchmark proposed in Section 3. Some of the considered methods require a specific training procedure consisting of several stages. Methods without Normalizing Flows, including **MLP**, **GNN**, **EnsGNN**, and **EN**, have only one training stage when the corresponding models train in the standard end-to-end mode. Methods with Normalizing Flows have at least three phases of training — warm-up of exclusively Flow neural layers, end-to-end training of the entire model, and finetuning the same Flow layers. In this regard, we follow the setup of Stadler et al. (2021). Further details can be found in Appendix G.

In our experiments, one training stage (i.e., warm-up, main training stage, or finetuning) takes 200 epochs, while the best loss value on the `Valid-In` part serves as a criterion for saving the model checkpoint. We exploit the standard Adam optimizer (Kingma & Ba, 2014) with a learning rate of 0.001 for Normalizing Flows and 0.0003 for other neural modules. For all the considered models, we utilize weight decay of 0.00001 and set $\lambda = 0.001$ in Expected Cross-Entropy.

As for model configurations, we set the hidden size of linear layers to 64, the number of layers in Normalizing Flows to 8 and the latent space dimension to 16. Also, Graph Propagation is performed in 5 steps with $\alpha = 0.2$.

