# OpenReview forum: "Revisiting Uncertainty Estimation for Node Classification: New Benchmark and Insights"
_ICLR.cc/2023/Conference — Submitted to ICLR 2023_

### Official Review · Reviewer_6NVp · 2022-10-24

**Confidence:** 4
**Correctness:** 2
**Technical Novelty And Significance:** 2
**Empirical Novelty And Significance:** 2
**Recommendation:** 6

**Clarity, Quality, Novelty And Reproducibility:**

This paper is well-presented and organized. This paper focuses on benchmark dataset construction. The authors provide an excellent code for experiment result reproduction.


**Strength And Weaknesses:**

Strengths:

1. This paper proposes a new benchmark for evaluating robustness and uncertainty estimation in transductive node classification tasks.
2. Using the proposed benchmark, the authors evaluate the robustness of various models and their ability to detect errors and OOD inputs


Weaknesses:

1. Similarly to the OOD detection in the image classification task, it is not necessary to use the Valid-out samples. I suggest conducting more experiments without Valid-out samples.
2. One important baseline (Zhao et al., 2020) is missing, which is introduced in Section 1. It is better to compare this baseline as well.
3. Since this paper focuses on the benchmark dataset, why not consider the MC-Dropout uncertainty estimation method as a baseline to compare?
4. For Table 1, what is Diff. %? Is that possible to construct different Diff. % for each dataset?


**Summary Of The Paper:**

This paper focuses on the Uncertainty Estimation for Node Classification problems and proposes a benchmark together with a technique for the controllable generation of data splits with various types of distributional shift. The proposed benchmark consists of several graph datasets equipped with various distributional shift on which we evaluate the robustness of models and uncertainty estimation performance.


**Summary Of The Review:**

See *Strength And Weaknesses*

------------ after rebuttal ----------------

The authors' response addressed most of my concerns. But I still suggest adding the MC-Dropout as a baseline in the future version as it is a benchmark paper. I will raise my score from 5 to 6.

---

> ### Author Response · Authors · 2022-11-08
> **Response to Reviewer 6NVp**
>
> Thank you very much for your feedback and suggestions! We reply to your comments below.
>
> > Similarly to the OOD detection in the image classification task, it is not necessary to use the Valid-out samples. I suggest conducting more experiments without Valid-out samples.
>
> Could you please explain why you think that the Valid-Out samples should be removed?
>
> In our benchmark, we create the Valid-Out part but do not use it in our experiments.
>
> We created this part to keep our benchmark consistent with previous works where the OOD validation parts exist. As mentioned in our paper, this part can be used for monitoring during training to see whether robustness can be improved when we have access to some OOD samples. Our results are reported in a more challenging scenario when the OOD validation part is not used.
>
> Additionally, the Valid-Out part allows us to create a gap between the ID subset, which represents the training domain, and the Test-Out part of the OOD subset, which serves for evaluating uncertainty estimation and robustness. This gap allows for a more challenging distribution shift, as it contains a range of samples (belonging to the Valid-Out part) that separate the observed ID subset, including Train, Valid-In, and Test-In, from the unknown Test-Out part in the OOD subset.
>
> > One important baseline (Zhao et al., 2020) is missing, which is introduced in Section 1. It is better to compare this baseline as well.
>
> As our main contribution is the benchmark for evaluating robustness and uncertainty, we examined a representative set of state-of-the-art algorithms. Zhao et al. (2020) have indeed made an important step towards uncertainty estimation in graphs, being one of the first to propose a Dirichlet-based framework. However, we decided not to consider this method since it was outperformed by GPN (Stadler et al., 2021), another Dirichlet-based method proposed specifically for node-level predictions, both in terms of misclassification detection and OOD detection.
>
> > Since this paper focuses on the benchmark dataset, why not consider the MC Dropout uncertainty estimation method as a baseline to compare?
>
> It has been shown on various problems and domains that the Monte-Carlo Dropout ensembling technique achieves consistently worse results compared to Deep Ensembles (e.g., see [1,2]), which in contrast are more diverse by construction and often provide very high performance. Because of that, we considered the experiments with MC Dropout not essential for our study and trained all the baselines without Dropout.
>
> Taking into account the number of datasets and split strategies, it would take much time and computational resources to provide the results for the MC Dropout method. However, we are ready to perform these experiments if you believe they could provide fundamentally new insights into robustness and uncertainty estimation for node classification. Please, let us know your opinion regarding this question.
>
> [1] Lakshminarayanan, Balaji, et al. “Simple and Scalable Predictive Uncertainty Estimation using Deep Ensembles”, NeurIPS, 2017.
>
> [2] Malinin, Andrey, “Uncertainty estimation in deep learning with application to spoken language assessment”, University of Cambridge, 2019.
>
> > For Table 1, what is Diff. %? Is that possible to construct different Diff. % for each dataset?
>
> The Diff. % values represent how much the node classification predictive performance decreases on the OOD test part compared to the ID one. It is computed as the difference between the accuracy scores on the OOD and ID parts divided by the accuracy score on the ID part. For an arbitrary method, this metric can be computed for each graph dataset and split strategy that induces distribution shift (e.g., we present this metric for the GNN baseline in Table 1). We will add an explanation to the caption of Table 1.

---

### Official Review · Reviewer_zVhc · 2022-10-25

**Confidence:** 4
**Clarity, Quality, Novelty And Reproducibility:** 1. The paper is easy to follow, excep…
**Correctness:** 3
**Technical Novelty And Significance:** 3
**Empirical Novelty And Significance:** 3
**Recommendation:** 5

**Strength And Weaknesses:**

Strength:

The proposed data-splitting approach accounts for the graph structures that existing benchmarks ignore. In the complex distribution shift scenario generated by the proposed approach, most existing methods fail to maintain a high classification performance and consistency of uncertainty estimates with prediction errors.

However, my major concerns are as follows.
1. The authors may want to provide references of realistic datasets which use the proposed PR and PPR splits. Otherwise, the proposed splits may amplify the locality-based and popularity-based biases in real-world applications.

2. Please add some large-scale datasets to the proposed benchmark. The seven datasets in the benchmark have only 2,000 to 35,000 nodes, but the graphs in real-world applications may be large-scale, e.g., ogbn-products [3] and ogbn-mag [3] have more than 1,000,000 nodes.
3. The authors may want to further analyze or discuss why the Dirichlet-based methods that the benchmark evaluates perform well or poorly on different tasks. This could provide insights for developing new Dirichlet-based methods.

4. The description of ${\rm PRC}\_{\rm oracle}$ needs improvements. In the case that uncertainties of all misclassified samples are different from each other, the value of ${\rm PRC}\_{\rm oracle}$ may not be unique. Which value of ${\rm PRC}\_{\rm oracle}$ do we choose to compute ${\rm PRR}$?

[3] Hu, Weihua, et al. "Open graph benchmark: Datasets for machine learning on graphs." Advances in neural information processing systems 33 (2020): 22118-22133.


**Summary Of The Paper:**

This paper proposes a benchmark for evaluating the uncertainty estimation in node classification tasks. The benchmark consists of seven common datasets, four problems for evaluating the performance of existing methods, the associated metrics, and a universal data-splitting approach. The key idea of the data-splitting approach is to use node characteristics depending on features or graph structures as splitting factors, leading to different distribution shifts. Experiments show that GPN [1] and NatPN [2] are the state-of-the-art methods for the OOD detection and the misclassification detection, respectively.

[1] Stadler, Maximilian, et al. "Graph posterior network: Bayesian predictive uncertainty for node classification." Advances in Neural Information Processing Systems 34 (2021): 18033-18048.
[2] Charpentier, Bertrand, et al. "Natural Posterior Network: Deep Bayesian Predictive Uncertainty for Exponential Family Distributions." International Conference on Learning Representations. 2021.


**Summary Of The Review:**

This paper proposes a benchmark for evaluating the uncertainty estimation in node classification tasks. The proposed new data-splitting method in the benchmark accounts for the graph structures that existing benchmarks ignore. In the complex distribution shift scenario generated by the approach, most existing methods fail to maintain a high classification performance and consistency of uncertainty estimates with prediction errors.

However, the paper is below the acceptance bar of ICLR because the proposed benchmark is not very helpful and insightful for real applications of the uncertainty estimation (please refer to concerns 1-3). Besides, there are some carless mistakes, e.g., Figures 6 and 7 are the same. If the authors can properly address my concerns, I am willing to raise my score.

---

> ### Author Response · Authors · 2022-11-08
> **Response to Reviewer zVhc**
>
> Thank you very much for your feedback and suggestions! We reply to your comments below.
>
> > The authors may want to provide references of realistic datasets which use the proposed PR and PPR splits.
>
> We aimed to make our splits diverse, realistic, and at the same time universally applicable to any dataset. Our PR-based split models situations when the most important elements are labeled first, which is realistic for many applications. For instance, in social networks, the most “important” users can be labeled first since they are the most influential. In citation networks, one may make conclusions based on the most cited publications. Similarly, in search engines, the most frequent queries are labeled first. However, when applying the model, we also face less popular items. Thus, it is important to be robust to such types of shifts. Our PPR split models a similar situation but is even more challenging. Here we add an aspect of locality — the labeled (in-domain) elements are localized in a specific region of the graph. This can also be natural: social networks are often explored by a local crawling starting from a particular node.
>
> > Otherwise, the proposed splits may amplify the locality-based and popularity-based biases in real-world applications.
>
> Let us emphasize that our work makes a step towards reducing such biases since we require the models to be robust towards such popularity-based shifts.
>
> > Please add some large-scale datasets to the proposed benchmark.
>
> The main motivation for choosing such graph datasets in our paper was to have a consistent comparison with the results of Stadler et al. (2021) since this work uses established graphs such as citation and co-purchase networks. Moreover, some works (e.g., [*]) study the necessity of large-scale graph benchmarks and suggest that diversity in terms of structural properties and joint distribution of features and graph structure might be more important for evaluating models than the dataset scale.
>
> Our approach to creating splits can be universally applied to any dataset, so we can produce the splits for the OGB datasets. However, evaluating all the models on these datasets would take significant time and computation resources. If you believe that these experiments are helpful and may provide fundamentally new insights, we will do our best to provide the results on these graph datasets.
>
> [*] Palowitch, John, et al. “GraphWorld: Fake Graphs Bring Real Insights for GNNs”, SIGKDD, 2022.
>
> > The description of $\text{PRC}\_{\text{oracle}}$ needs improvements. In the case that uncertainties of all misclassified samples are different from each other, the value of $\text{PRC}\_{\text{oracle}}$ may not be unique. Which value of $\text{PRC}\_{\text{oracle}}$ do we choose to compute $\text{PRR}$?
>
> Thank you for this comment; we will improve our explanation of computing PRR. Each model has its own oracle with the associated uncertainty estimates that perfectly match its prediction errors. So the $\text{PRC}\_{\text{oracle}}$ values of different models are different and computed only based on the corresponding model predictions.

---

> > ### Comment · Reviewer_zVhc · 2022-11-18
> > **Thanks for the authors' rebuttal.**
> >
> > Thanks for the authors' response. The response has addressed my concern 4.
> >
> > However, the authors' rebuttal does not properly address my concerns 1-3. Specifically, I suggest that the authors provide some references to support their claims and add some real-world large-scale datasets to the proposed benchmark in Concerns 1 and 2, while the aforementioned references and datasets are still missing. Besides, the authors do not respond to my concern 3.

---

> > > ### Author Response · Authors · 2022-11-18
> > > **Thanks for response**
> > >
> > > Thank you for replying to our revision.
> > >
> > > > Specifically, I suggest that the authors provide some references to support their claims and add some real-world large-scale datasets to the proposed benchmark in Concerns 1 and 2, while the aforementioned references and datasets are still missing.
> > >
> > > Regarding your concern 1, we provide a detailed discussion of our approach to creating distribution shifts in Section 3 and Appendix B. In particular, we discuss our motivation for considering various distribution shifts and describe how they can arise in practice. In our opinion, this motivation is quite reasonable and can be applied to many real-world graph machine learning problems. Moreover, if such publicly available benchmarks with real distribution shifts were already widespread, our work would not be needed. However, this is not the case, and creating meaningful shifts for evaluating robustness and uncertainty estimation is essential.
> > >
> > > As for the concern 2, we replied to this by asking for clarifications about the necessity of appending some large-scale datasets. All the reviewers asked for several experiments that require significant computation resources. Since we wanted to prioritize the tasks during the paper revision period, we asked the reviewers for advice. Unfortunately, they could not respond to us during the rebuttal period. Therefore, we decided to focus on experiments directly related to our main contribution, including a detailed analysis of distribution shifts and a comparison with other existing benchmarks.
> > >
> > > > Besides, the authors do not respond to my Concern 3.
> > >
> > > The in-depth investigation of Dirichlet-based methods is indeed an interesting problem. We made only a first step in this direction by comparing how different Dirichlet-based methods work under various distribution shifts. However, this problem is orthogonal to our main contribution since we suggest a new approach to creating data splits with meaningful distribution shifts rather than a new Dirichlet-based method for node-level uncertainty estimation.
> > >
> > > In general, we did our best to address the comments from all the reviewers, but additional large-scale experiments for all the models seemed out of the scope of our paper during the rebuttal period. Anyway, thank you very much for the time you spent reviewing our paper and your valuable feedback.

---

### Official Review · Reviewer_fvq7 · 2022-11-04

**Confidence:** 3
**Correctness:** 3
**Technical Novelty And Significance:** 2
**Empirical Novelty And Significance:** 1
**Recommendation:** 5

**Clarity, Quality, Novelty And Reproducibility:**

- I am confused about 40% of nodes being used as the OOD test set.
- I would like to see a comparison of Personalized PageRank with the shifts introduced in Gui et al. (2022).
- Furthermore, I would like to see datasets from different domains, especially molecular graphs.
- The code is available and it will be long-lived. However, code lacks documentation.

**Strength And Weaknesses:**

Strengths:
- Distribution shift problem is important to the graph learning community
- Various distributional shifts are considered
- Comprehensive experiments have been conducted to show that ID/OOD performance graph and SOA models performance.

 Weaknesses:
- I appreciate the extensive experimental efforts, however, the technical contribution of the paper is limited.
-  The paper does not spend much time discussing the distribution shifts which is the main contribution of the paper. I would like to see more discussion, motivation, and limitations of the Personalized PageRank (PPR).  Furthermore, the process of computing PPR is not explained well.
- The majority of the paper is dedicated to uncertainty estimation of nodes, metrics for evaluation, and methods which have been well documented in the literature and are not the main contribution here.

**Summary Of The Paper:**

This work presents a benchmark consisting of several graph datasets with different distributional shifts to evaluate the robustness and uncertainty estimation of models for node-level tasks.  Furthermore, a series of experiments have been conducted to show the ID/OOD performance gap.

**Summary Of The Review:**

In summary, this paper introduces distribution shifts in data splits, but the lack of technical novelty as well as evidence makes the current paper not strong enough for acceptance.

---

> ### Author Response · Authors · 2022-11-08
> **Response to Reviewer fvq7**
>
> Thank you very much for the feedback and suggestions! We reply to your comments below.
>
> > The paper does not spend much time discussing the distribution shifts which is the main contribution of the paper. I would like to see more discussion, motivation, and limitations of the Personalized PageRank (PPR).
>
> Thank you for this suggestion! In the revised version of the paper, we plan to provide additional analysis of the distribution shifts and data splits. Could you please be more specific regarding what you would like to see regarding an expanded discussion, motivation, and limitations of PPR so that we can address your request properly.
>
> > Furthermore, the process of computing PPR is not explained well.
>
> Thank you for this comment, we provide here a more detailed explanation of computing PPR and will also add it to our paper.
>
> First, we compute the standard PageRank score for every node in the considered graph. After that, we select a node $v$ with the highest value of PageRank (which can be thought of as the most important node in terms of the graph structure), and compute the Personalized version of PageRank for every node in the graph. Similarly to PageRank, PPR is a stationary distribution of a random walk on a graph, but restarts are always from the node $v$ with some probability $\alpha$.
>
> More formally, let $\mathbf{A}$ denote the adjacency matrix of the considered graph, $\mathbf{D}$ — the diagonal matrix of its degrees, and $\mathbf{e}_v$ — the one-hot-encoding vector of node $v$. Then, the PPR vector $\boldsymbol{\pi}$, each entry of which corresponds to some node, can be defined as follows:
> $$\boldsymbol{\pi} = (1 - \alpha)\mathbf{A}\mathbf{D}^{-1}\boldsymbol{\pi} + \alpha\mathbf{e}_v.$$
>
> These PRR scores can be treated as PageRank scores conditioned on the most important node in the graph. It means that nodes that are closer to the node $v$ will have higher scores.
>
> The main motivation behind the PPR-based splitting strategy is to create a locality-based split. PPR allows us to achieve this, as can be seen in our figures with visualization of data splits.
>
> > The majority of the paper is dedicated to uncertainty estimation of nodes, metrics for evaluation, and methods which have been well documented in the literature and are not the main contribution here.
>
> Do we understand correctly that you suggest improving the balance of the main text by expanding the parts related to our contribution with more analysis while shortening the parts describing standard concepts? We will try to achieve this in the revised version.
>
> > I am confused about 40% of nodes being used as the OOD test set.
>
> Could you please specify why this percentage is confusing?
>
> Our benchmark allows for flexible choice of the size of ID and OOD node subsets. This choice might depend on the practical needs and properties of a particular application. For our experiments, we decided to split the graph datasets into equal ID and OOD subsets (i.e., 50% each) and created an additional 10% gap between the actual OOD test part and the ID subset.
>
> > I would like to see a comparison of Personalized PageRank with the shifts introduced in Gui et al. (2022).
>
> Our work extends the contribution of Gui et al. (2022), taking into account not only node features but also graph structure, which we believe is important. In the revised version of the paper, we plan to compare our splits with Gui et al. (2022) by reporting statistics of the data splits. It would be very helpful if you could suggest some particular aspects of comparison with the GOOD benchmark that you would like to see so that we can better address this request.
>
> > Furthermore, I would like to see datasets from different domains, especially molecular graphs.
>
> Our work is focused on evaluating the robustness and predictive uncertainty of node classification models in the case of interdependent samples which occur in large graph datasets for node-level prediction. Concerning molecular graphs, most of the datasets are related to graph-level tasks, such as graph property prediction or link prediction, which is out of the scope of this work. If you have suggestions about suitable molecular graph datasets, we will try to add those during the author feedback period.
>
> > The code is available and it will be long-lived. However, code lacks documentation.
>
> Thank you for your positive feedback about our source code. We have updated the documentation in our repository.

---

### Author Response · Authors · 2022-11-17
**Revised paper**

Dear Reviewers, we have uploaded the revised version of our paper and did our best to address all the concerns.

**1. Motivation and analysis of the proposed distribution shifts.** In Section 3 and Appendix B, we provide a detailed discussion of our main contribution — the proposed distribution shifts. We focus on the motivation for choosing particular data split strategies, their relation to practical applications, and their quantitative comparison in terms of class balance, degree distribution, and pairwise shortest path distances. As a result, the discussion of distribution shifts has been extended in the main text, so we have moved some details about existing algorithms to Appendix E.

**2. Detailed comparison with GOOD (Gui et al., 2022).** In Appendix C, we provide a detailed comparison (both qualitative and quantitative) of our approach to creating distribution shifts to those proposed in GOOD (Gui et al., 2022). We explain why our work complements and extends GOOD, and what are the most important differences that make our benchmark more applicable in some aspects.

**3. Code documentation and text improvements.** We have also updated the documentation for our source code, made additional clarifications for some technical terms, and fixed several typos mentioned by the reviewers or found during the revision of our paper.

In summary, due to the valuable suggestions of Reviewers, we have managed to significantly improve our paper, make its content more clear, and better highlight our main contribution. We expect that our work now provides more insights into the performance of the considered methods and reveals what properties of our proposed distribution shifts make them important to consider in practice.

Please, let us know your opinion about the revised version of our paper.

---

### Decision · Program_Chairs · 2023-01-20

**Decision:**

Reject

**Justification For Why Not Higher Score:**

The limitation to small datasets, and the lack of deeper insights from the evaluation.

**Justification For Why Not Lower Score:**

N/A

**Metareview: Summary, Strengths And Weaknesses:**

This paper proposes a benchmark for evaluating the uncertainty estimation in node classification tasks. The benchmark consists of seven common datasets, four problems for evaluating the performance of existing methods, the associated metrics, and a universal data-splitting approach. The key idea of the data-splitting approach is to use node characteristics depending on features or graph structures as splitting factors, leading to different distribution shifts. Experiments rank different uncertainty estimation methods for OOD detection and misclassification detection.

Strengthes:
- it studies an important problem where there was a gap for evaluating uncertainty estimation under structural distribution shift
- it presents a novel idea of using PR and PPR to create data splits for structural distribution shifts

Weakness:
- the benchmark is limited in the size of the datasets. It is unclear the lessons learned from small datasets can be as informative for applications with substantially larger data size.
- while there is clear motivation to consider structural distribution shift for evaluation,  the motivation for the specific methods, PR and PPR, for creating the structural shift is unclear. There are many potential structural characteristics that can be used, some quite simple, like node degree or simple location based, why PR and PPR?
- the results are not surprising, seem to align with prior evaluation --- all in all, the results do not seem to reveal clear new insights